# Effects of Functional Phenolics Dietary Supplementation on Athletes’ Performance and Recovery: A Review

**DOI:** 10.3390/ijms23094652

**Published:** 2022-04-22

**Authors:** Ana C. Gonçalves, Dário Gaspar, José David Flores-Félix, Amílcar Falcão, Gilberto Alves, Luís R. Silva

**Affiliations:** 1CICS-UBI—Health Sciences Research Centre, University of Beira Interior, 6201-506 Covilhã, Portugal; anacarolinagoncalves@sapo.pt (A.C.G.); jdflores@usal.es (J.D.F.-F.); gilberto@fcsaude.ubi.pt (G.A.); 2CIBIT—Coimbra Institute for Biomedical Imaging and Translational Research, University of Coimbra, 3000-548 Coimbra, Portugal; acfalcao@ff.uc.pt; 3Department of Sport Sciences, University of Beira Interior, 6201-001 Covilhã, Portugal; dariogaspar7@gmail.com; 4Laboratory of Pharmacology, Faculty of Pharmacy, University of Coimbra, 3000-548 Coimbra, Portugal; 5CPIRN-UDI/IPG—Center of Potential and Innovation of Natural Resources, Research Unit for Inland Development (UDI), Polytechnic Institute of Guarda, 6300-559 Guarda, Portugal

**Keywords:** antioxidant properties, anti-inflammatory effects, phenolic compounds, exercise, recovery, sport performance, football, cycling, athletics

## Abstract

In recent years, many efforts have been made to identify micronutrients or nutritional strategies capable of preventing, or at least, attenuating, exercise-induced muscle damage and oxidative stress, and improving athlete performance. The reason is that most exercises induce various changes in mitochondria and cellular cytosol that lead to the generation of reactive species and free radicals whose accumulation can be harmful to human health. Among them, supplementation with phenolic compounds seems to be a promising approach since their chemical structure, composed of catechol, pyrogallol, and methoxy groups, gives them remarkable health-promoting properties, such as the ability to suppress inflammatory processes, counteract oxidative damage, boost the immune system, and thus, reduce muscle soreness and accelerate recovery. Phenolic compounds have also already been shown to be effective in improving temporal performance and reducing psychological stress and fatigue. Therefore, the aim of this review is to summarize and discuss the current knowledge on the effects of dietary phenolics on physical performance and recovery in athletes and sports practitioners. Overall, the reports show that phenolics exert important benefits on exercise-induced muscle damage as well as play a biological/physiological role in improving physical performance.

## 1. Introduction

In recent years, special attention has been paid to the incorporation of bioactive ingredients, especially from medicinal plants, in dietary supplements, nutraceuticals, and pharmaceuticals [1,2,3]. This is mainly based on the common belief that they are safer and more effective than synthetic drugs, and also due to their lower price, toxicity and side effects [4,5,6,7,8]. Among them, phenolic compounds have been a target of several studies and a hot topic for researchers. They constitute the largest group of secondary metabolites of plants, deriving from phenylpropanoids [1]. Their synthesis involves a series of deamination, hydroxylation, and methylation reactions that are mediated by the shikimate, where phenylpropanoids are formed, and by the acetic acid pathway, where the simplest phenolics are generated [9,10]. Due to their chemical structure, i.e., the presence of catechol, methoxy, and pyrogallol groups, they exhibit remarkable health-promoting properties, namely antioxidant, anti-inflammatory, antimutagenic, and vasodilator effects [11,12], which can be considered possible added values in sports’ performance and recovery [13,14,15].

It is undeniable the role of reactive species in human body. In fact, they act as signaling molecules, playing crucial roles in immune responses, regulation, expression, and signal transduction pathways [16,17]. However, and depending on the intensity, duration of the activity, training condition, age, and person gender, training status and genetic profile, in some situations, physical activities can lead to their overproduction and accumulation, shifting the oxidant/antioxidant ratio in favor of the oxidants, and therefore promote muscle soreness and severe ultrastructural damage [16,18,19,20,21]. Moreover, in the pre-seasons of most sports, many training sessions are carried out on the same day, leading to the increasing production of reactive oxygen and nitrogen species (RNOS) and pro-inflammatory markers [22,23,24]. For this reason, there is an intense search for ways to mitigate these effects and improve athletes’ performance and recovery. There is already evidence that dietary supplementation with phenolics or the regular intake of related dietary sources, such as cereals, fruits, herbs, spices, and vegetables, can offer several advantages given their ability to reduce oxidative stress, and consequently, reduce post-exercise inflammation and immune dysfunction, as well as improve metabolic recovery, aerobic capacity, muscle strength, and viral defense, thus having an interest in different sports with different physical demands [11,14,25,26,27]. Although the conjugation of phenolics with resistance training has not been adequately studied yet, it has already been shown that acute supplementation with approximately 500 mg of total flavonoids or 300 mg of procyanidins per day, 1–2 h before exercise, can effectively enhance athletic performance [28].

In view of the above, the purpose of this paper is to provide a detailed review of the effects of the major dietary sources of phenolics on the performance and recovery of athletes and sports practitioners. This overview is divided into three parts. The first two parts describe the major dietary phenolic sources, their bioavailability, and the relationship between their structure and health benefits. Finally, the third part consists of the latest findings and discussion on the main effects of phenolics in athletes and sports practitioners.

## 2. Data Collection

The authors of this review realized an exhaustive and detailed search for relevant articles regarding the potential of phenolics and phenolic-rich sources in sports. For that, data collection was conducted using different accepted scientific databases, including Google Scholar, Medline (PubMed), National Center for Biotechnology Information (NCBI), ResearchGate, Science Direct, Scopus, SpringerLink, Web of Science and other trusted journals publishes and abstracts until December 2021. The following free-terms, keywords or MeSH terms were applied: phenolics, phenolic-rich sources, phytochemicals, natural products, bioavailability, sports, recovery, performance, competition, antioxidant effects, anti-inflammatory properties, aerobic, anaerobic, demanding sports, athletics, exercises, eccentric exercise-induced muscle damage., and muscle soreness, and combining them with AND, OR, or NOT operators. No hold bars were imposed in terms of author(s) or type of publication during the literature survey. With the exception of one article (Mehri et al. [29]), the other ones were published in the English language. The almost 459 considered publications were carefully read, in order to find the more recent, complete and/or non-redundant publications meeting the objective of this work. In the final, there were selected 320 publications to be employed in this review.

## 3. Phenolic Compounds

Medicinal plants, along with cereals, fruits, herbs, and vegetables, are the main sources of phenolic compounds, of which nearly 10,000 have been reported in nature to date [30,31]. They are secondary metabolites produced by plants to protect them from abiotic factors (e.g., drought, extreme temperatures, floods, heavy metals, pH, radiation, salinity, and soils) and biotic factors, such as animals and pathogens attack [32]. In addition, they are considered to be mainly responsible for the organoleptic characteristics (e.g., aroma, flavour, and colour) of plants [33]. To facilitate their identification, they can be classified into five different subgroups based on their structure (at least one phenol ring), namely, (i) phenolic acids (which are further subdivided into hydroxybenzoic and hydroxycinnamic acids), (ii) flavonoids (subdivided into flavonols, flavan-3-ols, flavones, flavanones, isoflavones, flavanonols, and anthocyanidins), (iii) coumarins, (iv) lignans and (v) stilbenes (Figure 1) [34]. Among them, flavonoids are the most abundant in plants [35]. Flavonols are mostly found in apples, berries, grapes, broccoli, lettuce, onions, spinach, and tomatoes, while flavan-3-ols are mainly found in apples, bananas, blueberries, kiwi, peaches, pears, and their juices and jams, cereals, coca, broad beans, lentils, red wine and tea [36,37,38,39,40]. On the other hand, celery, tea, parsley, red pepper, and oranges contain significant amounts of flavones, while lemons and bitter oranges are rich in flavanones [37,38,39]. Isoflavones are mainly found in soy and soy products, lentils, beans and peas, and flavanonols, in lemons, grapes, oranges, and mint [36,37,38,40]. Finally, anthocyanins are largely identified in berry fruits and their juices and jams, pigmented cereals, purple corn, red cabbage, and red wine [36,40]. Regarding other phenolics, hydroxybenzoic acids have been reported in berries, onions, and black radish, whereas hydroxycinnamic acids are abundant in apples, berries, pears, plums, artichokes, carrots, chicory, eggplant, lettuce, wheat, and coffee [33,36,37,38]. Bison, cinnamon, lavender, sweet woodruff, and strawberries contain coumarins, while lignans are mostly found in sesame and flaxseeds [36,40,41]. Finally, stilbenes are predominant in blueberries, grapes, peanuts, and sorghum [42].

As mentioned earlier, their structure (Figure 1) allows them to readily neutralize free radicals and reactive species, interfere with signaling pathways, and reduce pro-inflammatory markers [4,32,43]. Therefore, it is not surprising that their scientific interest, consumption, and use in dietary supplements, functional foods, and pharmaceuticals are increasing worldwide [44]. However, although understanding the stability, solubility, bioavailability, and bioactivity of phenolics is not an easy task, it is essential because their health benefits depend mainly on the amount ingested and bioavailability [45]. In general, the bioavailability of different phenolics varies widely, and the most abundant phenolics in the diet are not necessarily the most active in vivo, either owing to their poor absorption by the intestine, rapid excretion, high metabolism, and/or lower intrinsic activity [38]. Nowadays, it is estimated that the absorption rates of the most consumed phenolics range from 2.7–12.2% for lignans, 8–72% for hydroxycinnamic acids, 33–100% for isoflavones, 12–41% for flavonols and flavones, 11–16% for flavanones, 2–8% for flavan-3-ols and 2.4–55% for anthocyanins [40,46,47,48,49,50]. Nevertheless, some studies reported that these percentages may eventually be higher [49,51]. In an attempt to improve the bioavailability and bioactivity of phenolic compounds, various efforts have been made and discussed. Among the alternatives, their encapsulation with carbohydrates (e.g., cellulose derivatives and maltodextrins), lipids (e.g., emulsifiers and waxes), natural gums (e.g., alginates and gum arabic), and/or with proteins (e.g., dairy proteins, gelatin, and soy proteins) seem to be the most promising strategies [45,52].

In addition, it is important to consider that bioavailability rates largely depend on the samples maturity stage, genotype, and agronomic conditions, which in turn, directly influence phenolic content, food processing, pH, solubility, chemical structure, and sugar moieties [50,53]. Furthermore, it is also important to consider interactions with other components present in food matrix, and the inter-individual variability between individuals (e.g., gender, age, food intolerances, genetic profile, enzymatic activity, pathological and/or physiological state of intestinal flora), lifestyle, and dietary habits [39,53].

Bioavailability of phenolics depends on different processes, including their release from the food matrix, gastrointestinal absorption, and presystemic metabolism; among them, intestinal absorption is considered the rate-limiting step and is the main culprit in reducing the bioavailability [35,54]. It begins in the mouth (Figure 2) with chewing, where the action of the oral microbiota initiates the metabolism of glycosylated compounds [48]. In the stomach, only a few compounds are hydrolyzed and deconjugated, with most remaining intact [55]. Here, some aglycones and simpler phenolic acids can be absorbed (bioaccessibility) [54]. The other pass to the small intestine, where cleavage and release of aglycones occur by the action of various digestive enzymes, with the activity of cytosolic *β*-glucosidase and lactase-phlorizin hydrolase being prominent [38]. Then, phenolics can be taken up by passive diffusion or transporters [55]. Some properties, such as lipophilicity, molecular weight, and glycosylation pattern, may affect their transport and permeability [35]. For example, isoflavones and gallic acid are readily absorbed due to their low molecular weight; on the other hand, proanthocyanins must first be degraded due to their high molecular weight [56,57]. The high lipophilicity of aglycones allows them to easily pass through epithelial cells by passive diffusion, whereas polymers, glycosides, and esters cannot pass through the membrane by passive diffusion and require transporters [56]. Quercetin glycosides, for example, require the action of sodium-dependent glucose cotransporter-1, while anthocyanins require glucose transporters 1 and 3 [35,58].

Nevertheless, the greatest absorption rate is observed in the colon [59]. Here, the gut microbiota catalyzes a series of reactions, including decarboxylation, demethylation, dihydroxylation, and hydrolysis, and transforms unabsorbed phenolics into bioavailable metabolites [55]. Usually, flavones and flavanones are degraded to hydroxy-phenylpropionic acids and polymeric phenolic compounds, while anthocyanins and proanthocyanidins are metabolizated to low molecular weight phenolic acids, flavonols converted into hydroxyphenylacetic acids, and flavanols into phenylvalerolactones and hydroxyphenylpropionic acids [52,60]. These acids are further metabolized to benzoic acid derivatives [60,61]. In the liver, they undergo some degree of phase I and/or phase II biotransformation reactions. Phase I includes hydrolysis, oxidation, and reduction reactions carried out by cytochrome P450 enzymes, while phase II aims to increase hydrophilicity before excretion [55]. Finally, metabolites may return to the small intestine via enterohepatic recirculation, be further absorbed by enterocytes, and return to the liver, or be eliminated [48].

## 4. The Linkage between Phenolics Structure and Health Benefits

It is clear from the literature, that phenolics have several beneficial effects on human health, including remarkable antioxidant, vasoactive and anti-inflammatory activities, and the ability to interact with enzymes, and cell receptors [62].

### 4.1. Antioxidant Effects of Phenolics

Regarding antioxidant mechanisms, different arrangements of functional groups in the core nuclear structure can confer metal chelation, radical scavenging, and/or oxidative activity to phenolics [63]. Although the role of free radicals and RNOS in human metabolism is undeniable, namely in the regulation of gene expression and cell growth, immune responses, and signal transduction pathways, their overproduction and consequent accumulation lead to DNA, lipids and proteins damage, necrosis, and useless inflammatory responses and promote the onset of several diseases, such as cancer, cardiovascular and neurological pathologies [4]. Unlike synthetic antioxidants, whose belief is that their constant intake is harmful and has undesirable effects, phenolics have already shown a notable ability to reduce oxidative stress levels with little or no negative effects [64].

Phenolics are effective in controlling oxidative stress and restoring the redox homeostasis due to their ability to (i) neutralize and/or reduce free radicals and the formation of reactive species, (ii) chelate trace elements involved in the formation of these pro-oxidant species, (iii) modulate the activity of related enzymes in cell signaling cascades, and (iv) stimulate the endogenous defense system by stimulating the action of intracellular antioxidant enzymes (e.g., catalase, glutathione, and superoxide dismutase) [34,65]. In this way, phenolics can decrease the production of pro-apoptotic multidomain effectors Bax and Bak protein, promoting an increase in the ratio of Bcl2–Bcl-xL/Bax–Bak, as well as stimulate the nuclear factor erythroid 2-related factor 2, and thereby, regulate the transcription of various antioxidant genes [66].

These effects are primarily attributed to their aromatic rings and the presence of several hydroxyl groups, which make them good hydrogen or electron donors [53,67]. However, the spatial position and the number of hydroxyl groups can affect their antioxidant potential [68]. As far as we know, phenolics with aromatic rings with two hydroxyl groups in the *ortho*-position (catechol group) are better antioxidants than those who only have aromatic rings with a single hydroxyl group (simple phenol groups) [34]. Procyanidin dimers followed by flavan-3-ols and flavanols are known to be the best antioxidants, followed by hydroxycinnamic acids and finally by simple phenolic acids [65].

As for phenolic acids, due to the CH=CH-COOH group and the 7,8-double bond in hydroxycinnamic acids, they are more efficient in reducing radicals and reactive species than hydroxybenzoic acids, which have only one COOH group (Figure 3A) [65]. In addition, the substitution of the carboxyl group with *O*-alkyl ester groups can also increase the antioxidant potential of phenolic acids, as well as the substitution with hydroxyl groups on the benzoic ring at the *ortho* and/or *para*-positions [69,70]. In contrast to *ortho*-methoxy groups, *ortho*-hydroxyl groups enhance these benefits [68]. On the other hand, the electron-withdrawing activity of the carboxyl group in phenolic acids reduces their capturing abilities [70]. Among phenolics, gallic and rosmarinic acids are considered to possess the highest antioxidant power (Trolox equivalent antioxidant capacity (TEAC) values of 3.62 and 4.5 mM, respectively) [65].

The antioxidant activity and free radicals scavenging ability of flavonoids are stronger than those of phenolic acids because they have a higher number of hydroxyl groups [63]. In fact, the presence of (A) an *o*-diphenolic group and 3′,4′-catechol hydroxyl groups on B ring, (B) a double bond in positions 2 and 3, and (C) the conjugation between the double-bond and the 4-oxo group on C ring enhance the antioxidant properties of flavonoids when compared to other phenolic subclasses (Figure 3B) [34,68]. Therefore, it is not surprising that the absence of any of these features abolishes the antioxidant activity of phenolics. The various hydroxyl groups on the B ring confer the capacity to give hydrogen atoms and electrons to hydroxyl, peroxyl, and peroxynitrite radicals, stabilizing them [68]. Moreover, the hydroxyl group in carbon 3 of the C ring increases the antioxidant power of flavonoids, standing out the quercetin activity (TEAC value of 4.7 mM), which has almost twice the antioxidant potential of catechin (TEAC value of 2.4 mM) [71]. Another clear example is the fact that although taxifolin and quercetin have a 4-oxo group, the absence of the double bond between carbons 2 and 3 of taxifolin, makes it a weaker antioxidant agent than quercetin [72]. In addition, the 5,7-*m*-dihydroxy arrangement on the A ring also increases the antioxidant capabilities of flavonoids [71].

All of these findings explain the lower antioxidant activity of kaempferol compared to luteolin; although both have the same hydroxyl configuration, kaempferol does not have the catechol group on the B ring, and, therefore, is less effective [73]. Moreover, the presence of a free hydroxyl group at position 3 in flavan-3-ols and flavonols is an added value because it can originate intramolecular hydrogen bonds with the 3′,4′-catechol structure of the B ring, aligning it towards the A ring [74]. This torsion angle on the B ring concerning the rest of the molecule contributes to the planarity of the structure, promotes conjugation and electron dislocations, and thus reduces radical species amounts [73]. A good example of this evidence is quercetin, which effectively presents remarkable chelating and scavenging potential, while flavones and flavanones are less effective in reducing oxidative stress because they do not possess a 3-hydroxyl group in their structure [75]. On the other hand, conjugation with carbohydrate and glycoside residues on A and/or B rings decreases the antioxidant activities of flavonoids [68]. This explains why quercetin aglycone has a stronger scavenging potential towards peroxyl radicals and ferric species than its *O*-glycosylated derivatives [71,76]. Among the sugar substitutions, the glycosylation at position 4′ is more suppressive than the substitutions at positions 3 and 7 of the C- and A-rings, respectively, whereas the substitution at position 3 elicits a higher antioxidant potential than that at position 7 [77]. Finally, it has also been reported that the polymerization of flavonoids also raises their antioxidant potential [78]. For this reason, procyanidin dimers and trimers are more effective than monomeric flavonoids in reducing free radicals and reactive species [79].

Relatively to stilbenes and lignans, the relationship between structure and biological activity is not completely understood. Until now, it is certainly that, in stilbenes, the hydroxylation at position 4 and the increasing number of hydroxyl groups at the *ortho* position are added values, conferring more solubility and facilitating interaction with proteins [80]. Concerning methoxylated groups, they confer resistance to degradation; nevertheless, an excessive number of them may difficult target interactions [81]. Regarding lignans, their antioxidant potential is positively associated with the presence of catechol (3,4-dihydroxy phenyl) residues and the butanediol structure but a higher degree of oxidation at the benzylic positions diminish their ability to scavenge free radicals and reactive species [82,83].

Furthermore, it is also important to consider the *O*-methylation of hydroxyl groups also influences the planarity of phenolics; particularly, the obstruction of the 3′,4′-catechol group by 4′-*O*-methylation significantly compromises their scavenging effects. A clear example happens with quercetin: its 4′-*O*-methylation decreases abruptly the capacity to avoid ferrous sulphate-induced lipid peroxidation [84]. Similar effects were observed with the kaempferol-3′,4′-dimethylether, where its capacity to scavenge peroxyl radicals is reduced to about half when compared to kaempferol [85].

However, it is important to keep in mind that phenolics’ chemical structure, including their catechol and pyrogallol groups, makes them susceptible to autoxidation [86]. When a phenolic loses an electron or acts as a reducing agent, and becomes a radical, although it can be stable, its oxidized intermediates can acquire a pro-oxidant behaviour [86,87]. For example, the interaction between phenolics and transition metals can lead to pro-oxidant events that, at high concentrations, can cause side effects to human health [68]. For instance, it is interesting to note that the glycosylation and methylation of hydroxyl groups of flavonoids can weaken the pro-oxidative behaviour [85]. Additionally, pH can also influence phenolics antioxidant capacity [88]. In general, a lower pH increases iron-reducing activity but inhibits iron catalytic activity and reduces chelate properties. For instance, at pH 7.4, *ρ*-hydroxybenzoic acid shows scavenging potential, while apigenin 7-glucoside presents a pro-oxidative behaviour. However, both compounds at pH 5.8 do not have any antioxidant or pro-oxidant activity [89].

Considering the preceding, it is extremely important to understand the relationship between the chemical structure and the activity of phenolics in order to avoid their auto-oxidation and find the most promising ones.

### 4.2. Anti-Inflammatory Properties of Phenolics

Inflammation is part of the complex biological response of body tissues to harmful stimuli caused by injury, infection, toxins, or rash [59]. During this process occurs the release of pro-inflammatory cytokines, such as tumor necrosis factor (TNF-*α*), interleukin (IL)-1*β* and IL-6, heat shock proteins, prostaglandins, and RNOS, including superoxide, nitric oxide, and hydrogen peroxide radicals [90]. When it is exaggerated, it triggers protein oxidation, damages carbohydrates, and nucleic acids and induces lipid peroxidation, and interferes with cellular functions, favouring the occurrence of several related ailments, such as metabolic problems, infections, rheumatoid arthritis, hypertension, and cancer [43]. Therefore, in uncontrolled situations, its removal is critical. Several efforts have been done to fully discover the anti-inflammatory capabilities of phenolics, being already reported that some of them can effectively relieve inflammatory responses, inhibiting expressions and/or activities of pro-inflammatory genes, including cyclooxygenases, angiotensin-converting enzyme, nuclear factor (NF)-κB, phosphoinositide 3-kinase/Akt (PI3 K/Akt) and mitogen-activated protein kinases (MAPKs) [66]. In addition, phenolics can activate p53, and hence, remove inflammation [91]. For example, *ρ*-coumaric acid showed a strong capacity to reduce TNF-*α* levels in a rat model of adjuvant-induced arthritis rats, whereas caffeic acid was able to inhibit lipopolysaccharide (LPS)-inducible nitric oxide synthase (iNOS), cyclooxygenase (COX)-2 expression, and TNF-*α* and IL-1*β* levels in RAW 264.7 macrophages [92,93]. Caffeic acid and ellagic acid together at 2% also demonstrated the capacity to mitigate the production of IL-1*β*, IL-6, and TNF-*α* in diabetic mice [94]. By focusing on flavonoid compounds, coloured and non-coloured phenolics extracted from sweet cherries revealed the capability to inhibit oxide nitric production (IC_50_ scores of 338.31 and 367.93 µg/mL, respectively) and to suppress iNOS and COX-2 expression in LPS-activated murine macrophage RAW 264.7 cells [43], while daidzein, isorhamnetin, genistein, kaempferol, naringenin, and quercetin showed the capacity to suppress NF-*κ*B and iNOS activation in murine J774 macrophages at 100 µM [90]. Kaempferol (0.02%) also revealed the ability to modulate the expression of NF-*κ*B genes in older rats [95].

## 5. Effects of Phenolic Compounds and Phenolic-Rich Sources on Exercise Performance and Recovery

Moderate physical activity is considered a key element in enhancing well-being and preventing or delaying the development of pathological ailments, mainly due to their ability to normalize reactive species levels [17,96]. Additionally, it also promotes several positive structural changes, including pathological adaptations regarding the cardiovascular system, and in brain regions related to motor planning, execution, learning and supervision [97,98,99,100,101,102]. However, in some situations, the overproduction and consequent accumulation of reactive species, mainly generated due to over-training and/or prolonged, unaccustomed, new and/or exhaustive sports and high intensity and/or competition sports may induce several aerobic and/or anaerobic bio-energetic reactions in cytosol and mitochondria [14]. This happens because some sports include explosive movements and/or eccentric contractions (e.g., downhill running, stepping, resistance training or cycling competitions) and/or extreme demands upon muscle (e.g., marathons, ultramarathons, and triathlons) and/or create a period of weakness and soreness that can last days or weeks (e.g., weightlifting) (Figure 4) [103,104,105,106]. Generally, inflammatory responses generated are controlled with the activation of neutrophils and macrophages and free radicals, namely nitric oxide, and, in the early stages, they are essential to create strength, promote muscle repair and regeneration, re-adaptation redox signaling pathways for the normal state, and optimize aerobic capacity, vasodilation, ventricular function, insulin resistance, and hyperemia during exercise [106,107,108,109]. Additionally, the hydrogen peroxide release, through oxidation and activation of NF-*κ*B (IκB) kinases can regulate the NF-*κ*B pathway, which, in turn, can play a protective role under conditions of oxidative stress, by avoiding the accumulation of reactive species [110]. Nevertheless, when there is an imbalance (e.g., muscular catabolism, fiber damage, impairment of neuromuscular functions, training adaptations, among others), immunosuppressive events, extreme fatigue, and excessive inflammatory reactions may occur, which, in turn, can become harmful for human health, and limit the performance in competitions [111]. Particularly, the overproduction of superoxide radicals derived from inflammation leads to their reaction with nitric oxide, forming highly reactive oxidant peroxynitrite species [112]. Similarly, superoxide radical can also interact with the protons from lactic acid, being hence converted into perhydroxide species, which in turn, also possess high damage [21,113]. However, it is important to take into account that such side effects are also dependent on the intensity and duration of the activity and the training status of each individual [15].

With this in mind, several studies have been conducted to prevent or at least mitigate structural muscle damage, oxidative stress, peroxidation, and the release of Ca^2+^-activated proteases, and in this way, ameliorate contractile dysfunction and evoke a better recuperation [114]. Among them, the intake of phenolic compounds appears to be a beneficial strategy thanks to their ability to suppress inflammation by inhibiting oxidative damage and interacting with the immune system, and in this way, recover muscle strength and soreness, reduce fatigue and cellular damage, and preserve myocytes (Figure 5) [105,115]. In addition, they can also contribute to improving physical performance, endurance, recovery, and cardiovascular function during exercise, as well as contributing to better psychological and physiological conditions of the athletic community [114,116].

Even so, and although some experts alert that excessive amounts of phenolics can be dangerous given the absence of studies and long-term health benefits, dietary supplementation of botanical origin is increasing worldwide, which is not surprising considering their nature and medicinal use since ancient times in folk medicine [117]. Furthermore, there is also observed a growing interest in those related to sports and supplemented with vitamins and minerals. Focusing on these micronutrients, vitamin C and E supplements have been described as a promising strategy to reduce, or even prevent, damage in active muscles and increase the antioxidant capacity. These effects were already observed in individuals who ingested 1000 mg of vitamin C for 2 weeks, 2.5 h before cycling [118]. On the other hand, athletes who daily intake 400 mg of vitamin C for 2 weeks before a 90-min intermittent shuttle running test showed lower concentrations of IL-6 and malondialdehyde in plasma [111]. Moreover, young men who ingested 800 IU of *α*-tocopherol 48 days before downhill running, displayed reduced IL-1*β* and IL-6 levels, while the 14-day and 30-day daily ingestion with 1200 IU *α*-tocopherol are effective in decreasing creatine kinase (CK) plasma and serum amounts, respectively, after different exercises practices [104,119]. In line with the mentioned, a nutrigenomic approach related that the daily consumption by humans of a mix of resveratrol, green tea extract, *α*-tocopherol, vitamin C, polyunsaturated fatty acids, and tomato extract during 5 weeks can effectively reduce the inflammation in adipose tissue, improve endothelial function, increase liver fatty acid oxidation and reduce oxidative stress levels [120].

In terms of phenolics, evidence already indicated that the supplementation with approximately 300 mg of phenolics 1–2 h prior to the activity can improve exercise capacity and performance during endurance and repeated sprint workout [15,28], whereas, the supplementation with more than 1000 mg of phenolics per day for 3 or more days before and after exercise can improve recovery after muscle injury [107,121]. These positive effects occur owing to phenolics capacity to modulate antioxidant and anti-inflammatory pathways by inhibiting the activity of the main source of superoxide production during exercise, which is NADPH oxidase, and markers of inflammation (e.g., C-reactive protein (CRP), TNF-*α*, and IL-1*β* and 6 levels) and muscle damage (e.g., lactic dehydrogenase (LDH), myoglobin, CK, urea, uric acid, alanine aminotransferase, and aspartate aminotransferase concentrations on bloodstream), and hence, improving oxidant defenses and reducing radicals levels [105,106].

Table 1 and Table 2 summarize the most important in vivo assays in rodent models and also the most notable clinical data reported to date based on the ability of phenolics and phenolic-rich sources to improve athletes’ performance and recovery. Even so, it is important to consider that deeper studies are needed to full reveal their biological potential and safe dosage, even because, it had been already reported that some phenolics, at high amounts, can exhibit carcinogenic and/or genotoxic effects, become prooxidant, interact with some pharmaceutical drugs, or may interfere with thyroid hormone biosynthesis, promoting unwanted side effects [122,123]. Given that, it is not surprising that, in some situations, will be recommended a diet poor in phenolics [63]. Particularly, Gonçalves et al. [43] reported recently that anthocyanin-rich fractions at concentrations above 100 µg/mL from sweet cherries revealed pro-oxidant behaviours and did not show potential to protect neuroblastoma SH-S5Y5 cells against hydrogen peroxide-induced oxidative stress. Similar effects were attributed to quercetin (10–100 µM) in melanoma B16F-10 cells and in the myeloperoxidase-rich human HL-60 cell lines [124]. Catechins from green tea (0.1% or 1% of the diet) enhanced tumour development in the colon of F344 male rats [125]. In addition, it was also reported that the daily administration of caffeic acid at amounts of 678 and 814 mg/kg/day by rats, and 2120 and 3126 mg/kg/day by mice induce in both rodents forestomach and kidney tumours [126]. Additionally, high intake of isoflavones (at values above 120 mg isoflavones per day considering human doses) reduced fertility and generate liver diseases in animals, exhibited anti-luteinizing hormone effects in premenopausal women and delay sexual maturation of infants [127,128,129,130,131,132]. Given the mentioned, it is important to consider the metabolism and genetics of each individual.

### 5.1. Phenolic Compounds

As can be expected, among phenolics, quercetin, although it demonstrated pro-oxidant behaviours in specific situations [124], is one of the most studied because of its abundance in the human diet and powerful antioxidant properties. It is largely found in apples, berry fruits, red grapes, onions, leafy green vegetables, hot peppers, and black tea [133]. In vitro assays have already verified that quercetin at concentrations varying from 0.3 to 30 µM can attenuate the expression of the main contributors to muscle damage, i.e., suppress TNF-*α*, IL-1*β,* IL-6 and interferon-γ levels in human macrophages [134]. Resveratrol is another phenolic that has been extensively studied. It is found primarily in seeds and skins grapes, mulberries, peanuts, rhubarb, and red wine [38]. Based on in vitro assays, this compound showed the ability to upregulated mitochondrial biogenesis factors SIRT1 and PGC-1*α* in human coronary arterial endothelial cells at 10 µM, after 24 h of exposure [135]. Moreover, it also showed the capability to promote cardioprotective effects on endothelial cells and cultured rat aorta, by decreasing oxidized low-density lipoprotein and TNF-*α*, glutathione peroxidase, and heme oxygenase-1 and providing protection against oxidative damage by hydrogen peroxide, at doses ranging from 1 to 100 µM [136]. These results are in agreement with data obtained in vivo with rodents [137,138].

#### 5.1.1. Rodent Models

Relatively to in vivo assays, Davis et al. [139] reported that a 7-day quercetin supplementation (12.5 and 25 mg/kg) in mice can improve running time to fatigue by increasing mRNA expression of transcriptional coactivators sirtuin 1 (SIRT1) and peroxisome proliferator-activated receptor-*α* coactivator (PGC-1*α*), and also of mitochondrial DNA and cytochrome C in mice. It has also been reported that supplementation of quercetin for 7 days (12.5 mg/kg/day) can reduce the risk of influenza infection in rats subjected to intensive runs.

As already mentioned, resveratrol has also been a target of several studies. In fact, Rimbaud and coworkers [140] have shown that 12-week treatment with resveratrol (20 mg/kg/day) in rats with induced hypertension can preserve mitochondrial mass and biogenesis and increase their survival rate, essentially by increasing the expression of peroxisome proliferator-activated receptor (PPAR)-*α* and counteracting the development of cardiac dysfunction and fatty acid metabolism. Similar data were reported by Esfandiarei and colleagues [141] and Broderick et al. [142] who reported the daily supplementation of resveratrol at 4 g/kg along with regular exercise training can improve the aortic morphology of the 3xTg mouse model of Alzheimer’s disease, as well as increase neurotrophins, synaptic markers, and silent information regulators and reduce neuroinflammation, accumulation of A*β*-oligomers, and markers of apoptosis, autophagy, endolysosomal degradation, and ubiquitination events. In addition, resveratrol supplementation at 10 mg/kg/day has been documented to be able to reduce lipid peroxidation by lowering malondialdehyde and 8-hydroxy-2′-deoxyguanosine levels in rats subjected to intensive swimming [143], as well as to improve recognition memory and anxiety-like behaviours, and regulate muscle glycogen [144,145]. Furthermore, Guo and coworkers [146] demonstrated that 9-week supplementation (50 mg/kg/day) improved reproductive function in rats subjected to high-intensity workouts by increasing sperm density, testosterone, and follicle-stimulating hormone levels, protamine, superoxide dismutase activity, and spermatogenic epithelial cell number, and reducing inflammatory markers (IL-6, TNF-*α*, and malondialdehyde content). On the other hand, resveratrol supplementation (10 mg/kg/day) for 4 weeks exerted a regulatory effect on serum iron and magnesium levels in rats’ blood undergoing acute swimming exercises [147]. Moreover, an 8-week supplementation at 20 mg/kg in diabetic rats exposed to regular continuous exercise increased B-cell lymphoma 2 (BCL-2) levels, and reduced Bcl-2-associated X-protein (BAX), caspase 3 [148], and hepatocyte apoptosis [29]. It was also found that its supplementation (50 mg/kg/day) in rats 6 h before they were subjected to intense swimming was effective in restoring LDH, ammonia, creatine phosphokinase (CPK), and glucose levels at rest [149]. Moreover, a 6-week supplementation (7.5 mg/kg/day) exerted cardiovascular protective effects by upregulating the expression of phosphodiesterase (PDE) 1, 2, 3, and cAMP-elective PDE4 in male and female rats subjected to regular aerobic exercise [150].

Additionally, salidroside, a simple phenol, also shows potential to be considered as a promising strategy. Indeed, Li and coworkers [151] reported that salidroside supplementation (50 mL/kg/day) can increase superoxide dismutase and glutathione peroxidase activities, and free fatty acid concentrations in liver glycogen and muscle glycogen reserves, and decrease malondialdehyde levels in mice forced to swimming exercises for 2 weeks.

#### 5.1.2. Clinical Studies

Based on quercetin daily consumption of approximately 13.58 mg/day, many studies have been conducted to assess its full potential, including its effects on athletes [152]. Although exist some clinical trials that did not report significant changes or ergogenic effects after quercetin supplementation [76,153,154,155,156,157,158,159], a recent report revealed that cyclists who daily intake quercetin for 6 weeks (300 mg/day) showed better aerobic performance (+3.1% regarding the 30 Km trial and by +2% over the last 5 km) when compared to placebo [160]. In addition, 1000 mg/day of its supplementation for 3 weeks before, during, and 2 weeks after an intense 3-days of exercise may reduce the risk of upper respiratory tract infection in trained male cyclists [133]. It has also been reported that a 2-week supplementation can increase isometric strength and decrease the severity of muscle weakness caused by eccentrically induced myofibrillar dysfunction and impaired sarcolemmal action potential propagation [161]. In addition, a meta-analysis performed by Pelletier and colleagues [162] found that quercetin intake increased endurance performance by 0.74% in trained and untrained individuals when compared with the untreated group. Moreover, Davis et al. [163] reported that a 1-week quercetin supplementation (1000 mg/day) in untrained males can increase their maximal oxygen consumption (VO_2max_) and time to fatigue. On the other hand, the combination of 500 mg quercetin and 250 mg vitamin C exerted remarkable anti-inflammatory effects and oxidative protection, namely by reducing CRP, IL-6, E-selectin, and F2-isoprostane levels [160]. Overall, the beneficial effects of quercetin are mainly attributed to its ability to cross membranes and promote their stability, thus maintaining excitation-contraction coupling in myocytes and strengthening the immune system [161]. The absence of more positive data may be related to the timing of the report, the doses studied, and also to the redox status and interindividual variability among participants. Nonetheless, there is evidence that prolonged intake of quercetin adds value [156,159,164].

As quercetin, resveratrol also shows a promising effects in athletic men and women [165,166,167,168]. Particularly, Eschle and colleagues [169] reported that daily supplementation of 500 mg resveratrol can significantly increase respiratory exchange by improving carbohydrate oxidation during cognitive demands. It was also reported that its supplementation (3 g/day) for 12 weeks increased skeletal muscle SIRT1 expression and energy expenditure by upregulating AMPK expression in patients with type 2 diabetes mellitus [170]. Moreover, it was also reported that its combination at 25 mg/kg with nonsteroidal anti-inflammatory drugs showed the ability to accelerate muscle recovery in rats subjected to contusion, by decreasing serum creatinine (CREA), CK, LDH, and uric acid levels after 7 days of supplementation [171]. On the other hand, the daily intake of 500 mg resveratrol plus 10 mg piperine, an alkaloid responsible for the pungency of black and long pepper, increased forearm skeletal muscle mitochondrial capacity in young adults who performed wrist flexor workouts for 4 weeks [172]. In addition, McAnulty et al. [173] reported that supplementation of quercetin with resveratrol (120 mg resveratrol and 225 mg quercetin for 6 days and 240 mg resveratrol and 450 mg quercetin on day 7 just before exercise) was effective in reducing F2 isoprostane levels and lipid peroxidation. More recently, Kawamura et al. [174] conducted a 10-week study and confirmed that supplementation with astaxanthin, *β*-carotene, and resveratrol-rich foods improved resistance training-induced strength, metabolic adaptations, and reduced fatigue, and oxidative damage. However, the use of resveratrol needs to be more explored since some pro-oxidant behaviours and cytotoxic effects in normal cell lines have been related [175].

Regarding ellagitannins, which are mainly found in almonds, black raspberries, pomegranates, raspberries, strawberries, and walnuts, it was verified that their 500 mL daily intake over 9 days is effective in increasing strength after eccentric exercise in active men [176]. On the other hand, a 4-week supplementation of hesperetin, a flavanone found mainly in citrus fruits, at a dosage of 500 mg/day, showed the ability to increase strength in trained cyclists’ males [177].

Flavan-3-ols, namely catechins and epicatechins, also appear to be a useful tool in sports. In particular, Jajtner et al. [178] reported that a 7-day supplementation of 2000 mg/day of catechins can effectively reduce IL-8 levels in untrained men undergoing resistance training. In addition, Kim et al. [179] demonstrated that a 3-months supplementation (350 mL of a tea drink fortified with 540 mg of catechins) increased leg muscle mass, and walking speed in elderly women with sarcopenia who exercised twice a week for 60 min. These effects are closely related to the ability of flavan-3-ols to upregulate the activity of catechol-*O*-methyltransferase in the bloodstream, thus enhancing antioxidant activity [180,181,182]. On the other hand, the supplementation with epicatechin (1 mg/kg/day) for 8 weeks showed the ability to increase the follistatin/myostatin ratio, and therefore, improve muscle strength [183]. Concerning its derivative epigallocatechin 3-*O*-gallate, its supplementation at a dose of 135 mg/day and 150 mg twice daily increased VO_2max_ in healthy subjects before specific exercise test, and without affecting maximal cardiac output, it can decrease heart rate and plasma glucose in obese postmenopausal women who underwent to regular aerobic exercise [184,185]. In addition, the daily intake of 1800 mg in combination with *N*-acetyl-cysteine, a potent synthetic antioxidant, can diminish neutrophils count, and CK, 8-isoprostane, cortisol, and TNF-*α* levels [186].

The intake of anthocyanins-enriched juice (240 mL twice daily) for a period of 9 days also showed beneficial effects, being able to lower CK levels and shorten the recovery time [187]. These effects are mainly attributed to the notable antioxidant scavenging potential and consequent ability to suppress uncontrolled pro-inflammatory responses of anthocyanins [43,53].

On the other hand, clinical studies have shown that isoflavones at a dose of 70 mg/day are truly effective in reducing fat mass, plasma γ-glutamyl-transferase, and CRP levels when combined with 6 months of exercise [188,189,190]. Moreover, the flavanone hesperetin at a dosage of 500 mg/day for a period of 4-weeks showed potential to increase absolute power and decrease VO_2max_ in male athletes undergoing a cycling training program [177].

Finally, curcumin has also been the subject of numerous studies, primarily for its anti-inflammatory effects [191]. With regards to its effects on exercise, curcumin has already shown efficacy in reducing eccentric muscle soreness and oxidative stress markers, as well as in attenuating acute inflammation at doses above 150 mg/day [106,191,192,193,194,195].

As for phenolic acids, caffeic acid at doses of 1, 2 and 4 µg/mL showed efficacy in reducing hyperthermia-induced survival inhibition, necrosis, superoxide levels, and glutathione depletion in mononuclear cells of cyclists exposed to hyperthermic stress [196]. On the other hand, in rats exposed to eccentric exercise, a 5-day supplementation at doses of 5 and 10 mg/kg/day was effective in downregulating NF-*κ*B, and consequently, lowering COX2, iNOS, IL-1*β*, and MCP1 levels [197].

In a general way, the absence of more positive results in clinical trials is mainly due to the short period of time and/or the doses chosen. In addition, more rigorous and controlled studies are needed for better conclusions.

**Table 1 ijms-23-04652-t001:** Main effects of phenolic compounds in exercise.

Phenolic Compound	Study Design	Main Outcomes	Reference
Rodent Models			
Catechin			
	Effects of 8-week supplementation (0.35% catechins/day) in mice subjected to downhill exercise	Attenuated the downhill running-induced decrease in muscle force ↓↓ pro-inflammatory markers in both plasma and gastrocnemius muscle ↓↓ oxidative stress, by ↑↑ glutathione reductase activity	[181]
(−)-Epigallocatechin-3-*O*-gallate			
	Effects of 16-week supplementation (0.32%/day) in the skeletal muscle of high fat-fed mice	↓↓ fasting blood glucose (−18.5%), plasma insulin (−25.3%), and insulin resistance (−33.9%), and markers of obesity-related fatty liver disease in high fat-fed mice ↑↑ mRNA levels of the nuclear respiratory factor, medium-chain acyl-CoA decarboxylase, uncoupling protein 3, and peroxisome proliferator responsive element	[198]
Quercetin			
	Effects of 7-day supplementation (12.5 mg/kg/day) against influenza infection before an intensive run in rats-model	Quercetin administration effectively ↓↓ the susceptibility to influenza infection following stressful exercise	[199]
	Effects on brain and muscle mitochondrial biogenesis and exercise tolerance in rats at doses of 12.5 and 25 mg/kg	↑↑ mRNA expression of PGC-1*α*, SIRT1, mtDNA and cytochrome C ↑↑ maximal endurance capacity and voluntary wheel-running activity	[139]
Resveratrol			
	Effects of 4-week supplementation (10 mg/kg/day) in rats subjected to an acute swimming exercise bout for 30 min	↓↓ lipid peroxidation and genotoxicity, by diminishing MDA and 8-OhdG levels ↑↑ total antioxidant levels	[143]
	Effects of 12-week supplementation (146 mg/kg/day) in rats subjected to a progressive treadmill running	↑↑ exercise training-induced improvements (+21% than placebo) ↑↑ twitch and tetanic forces generated during isometric contraction (+18 and 58%, respectively) and cardiac fatty acid oxidation (+20% than placebo) Favourable changes in cardiac gene expression and structure, and signal transduction pathways	[200]
	Effects of 4-week supplementation (10 mg/kg/day) in rats subjected to an acute swimming exercise bout for 30 min	Prevented the decrease of glycogen in the liver after intense exercise in both exercised and non-exercised rats	[201]
	Effects of 12-week supplementation (10 mg/kg/day) in rats subjected to a treadmill endurance exercise program	Prevented exercise-induced oxidative stress by avoiding lipid peroxidation (lower levels of MDA and higher superoxide dismutase activity and total antioxidant capacity on plasma)	[202]
	Effects of 6-week supplementation (7.5 mg/kg/day) in male and female rats subjected to regular aerobic exercise	↑↑ antioxidant capacity and the expression of non-selective PDE1, 2, 3 and cAMP selective PDE4 There were not observed effects on cGMP selective PDE5 expression in the aorta	[150]
	Effects of one-year supplementation (0.7 mg/kg/day) on skeletal muscle of active old mice	↑↑ antioxidant defences, blocking SkM protein carbonylation increments ↓↓ IL-6	[203]
	Effects of 6-week supplementation (10 mg/kg/day) in aged rats subjected to swimming high-intensity interval exercise	↑↑ recognition memory and modulate anxiety-like behaviours	[145]
	Effects of 4-week supplementation (10 mg/kg/day) in rats subjected to acute swimming exercise	Protective effect on muscle glycogen in exercised rats	[144]
	Effects of 4-week supplementation (25 mg/kg) in skeletal muscle adaptation involved in exercise-induced weight loss in obese mice	↑↑ whole-body glucose, lipid homeostasis and the expression levels of PGC-1*α* and its downstream transcription factors	[204]
	Effects of 2-week supplementation (12.5 mg/kg) in rats’ gastrocnemius muscles	Protective effects against oxidative stress and muscle force loss in hindlimb suspension ↓↓ hydrogen peroxide, lipid peroxidation, and caspase-9 levels in muscles of animals after hindlimb suspension subjected to hindlimb suspension	[205]
	Effects of 4-week supplementation (25, 50, and 100 mg/kg) in rats subjected to a strenuous exercise on the treadmill	↓↓ lipid peroxidation by diminishing LDH, CK, MDA, 4-HNE, and 8-OHdG levels	[206]
	Effects of 8-week supplementation (20 mg/kg) in diabetic rats subjected to continuous exercise	↓↓ hepatocyte apoptosis caused by diabetes	[29]
	Effects of 10-day supplementation in young adult and aged mice gastrocnemius muscles subjected to short-term isometric exercise	↓↓ oxidative stress and oxidative damage in gastrocnemius muscles from young adult and aged mice subjected to short-term isometric exercise	[207]
	Effects of 8-week supplementation (20 mg/kg) in diabetic rats subjected to regular continuous exercise	↓↓ BAX and caspase-3 ↑↑ BCL-2 concentration in hepatocyte tissue	[148]
	Effects of 6-month supplementation (16.5 mg/kg/day) in rats subject to running exercises	↑↑ antioxidant defenses by increasing glutathione, glutathione peroxidase, glutathione transferase, and NAD(P)H: quinone acceptor oxidoreductase activities	[208]
	Effects of 9-week supplementation (50 mg/kg/day) in rats’ spermatogenic dysfunction caused by high-intensity exercise	↑↑ significantly sperm density, testosterone and FSH levels, protamine, superoxide dismutase activity, and spermatogenic epithelial cells number ↓↓ IL-6, TNF-*α* and MDA content	[146]
	Effects of supplementation (50 mg/kg/day) in rats 6 h before intensive swimming	↑↑ accelerated the recovery of LDH, ammonia, CPK, and glucose levels after exercise	[149]
	Effects of 5-month supplementation (4 g/kg/day) in induced-Alzheimer’s disease rats subjected to the treadmill belt	↓↓ neuroinflammation, accumulation of A*β* oligomers and markers of apoptosis, autophagy, endolysosomal degradation, and ubiquitination ↑↑ increased levels of neurotrophins, synaptic markers, and silent information regulator	[142]
	Effects of 4-week supplementation (10 mg/kg/day) in rats subjected to an endurance exercise and acute exercise training	↓↓ carbonyl and 8-OHdG levels	[209]
	Effects of 12-week supplementation (10 mg/kg/day) in rats subjected to endurance exercises	↓↓ CPR and IL-6 levels No effects regarding TNF-*α* and IL-17 content were observed	[210]
	Effects of training and resveratrol 12-week supplementation (100 mg/kg) on the muscle of colon cancer mice	↑↑ muscle regeneration by increasing Myosin heavy chain-embryonic protein (+34.7% than placebo) No effects regarding myoblast determination protein, body weight and tumour size were observed	[211]
	Effects of training and resveratrol 3-week supplementation (25, 50, and 125 mg/kg/day) in rats subjected to forelimb grip strength and exhaustive swimming	↓↓ serum LDH, ammonia, and CK levels in a dose-dependent manner ↑↑ grip strength and glucose levels in a dose-dependent manner	[212]
	Effects of 4-week supplementation (25 mg/kg/day) in rats subjected to forelimb grip strength exercises	↓↓ plasma LDH and ammonia levels ↑↑ muscle strength and endurance performance	[213]
	Effects of training and 6-week supplementation (40 mg/kg/day) in rats subjected to swimming	↓↓ kainate-induced seizure activity and oxidative stress, by increasing superoxide dismutase activity	[214]
	Effects of training and 6-month supplementation (16.67 mg/kg/day) in rats	↑↑ catalase, superoxide dismutase, glutathione peroxidase, glutathione reductase, glutathione S-transferase and NAD(P)H: quinone oxidoreductase 1	[215]
	Effects of training and 4-month supplementation (4 g/kg/day) in Alzheimer’s disease model	↑↑ improves fracture resistance and cross-sectional geometric indicators of bone strength	[216]
	Effects of 6-week supplementation (16–17 mg/kg/day) in old mice	↓↓ frailty ↑↑ muscle performance, coordination and strength practising a moderate exercise	[217]
	Effects of training and 6-week supplementation (150 mg/kg/day) in rats subjected to running	Strong effects on weight loss, which could be beneficial for suppressing exercise loads	[218]
	Effects of training and 9-week supplementation (100 mg/kg/day) in rats subjected to running	↑↑ PGC-1*α* and SirT1 to the nucleus, stimulating mitochondrial biogenesis	[138]
	Effects of training and 4-week supplementation (15 mg/kg/day) in rats subjected to swimming	↑↑ PGC-1*α*, and hence, stimulating mitochondrial biogenesis, and also citrate synthase enzyme activity	[219]
	Effects of training and 13-week supplementation (0.2% w/w) in rats subjected to running	↑↑ senescence-accelerated prone mice levels Improve energy metabolism in skeletal muscle	[137]
	Effects of resistance exercise and 4-week supplementation (25 mg/kg) in rats subjected to aerobic and anaerobic exercises	↑↑ muscular hypertrophy, physiological adaption, and aerobic and anaerobic performances	[220]
	Effects of 12-week supplementation (100 mg/kg) in rats subjected to treadmill exercise training	↑↑ aerobic performance, mitochondrial quality, control, and biogenesis, namely by activating the AMPK-SIRT1-PGC-1*α* pathway	[221]
	Effects of 8-week supplementation (10 mg/kg) in rats subjected to forced running	Better exhausted time and running distance	[222]
	Effects of training and 8-week supplementation (25 mg/kg/day) in the liver of elderly rats with nonalcoholic fatty liver	↑↑ Sirt1, Lxr, and Fxr, ↓↓ AST, ALT, ALP enzymes, and apoptotic cells	[223]
Resveratrol conjugated with nonsteroidal anti-inflammatory drugs			
	Effects of 7-day supplementation (25 mg/kg) in rats after contusion induced muscle injury	↑↑ muscle recovery, by lowering uric acid, creatinine, LDH, and CK serum levels	[171]
Caffeic acid phenethyl ester			
	Effects of 5-day supplementation (5 and 10 mg/kg/day) in rats subjected to eccentric exercises	↓↓ COX2, iNOS, and IL-1*β* and MCP1 levels, by suppressing NF-*κ*B activation	[197]
Salidroside from *Rhodiola Sachalinensis* A. Bor			
	Effects of 2-week supplementation (50 mL/kg/day) in mice forced to swim for 120 min without loads on the last day of the assay	↑↑ superoxide dismutase and glutathione peroxidase activities, and liver glycogen and muscle glycogen reserve and free fatty acid concentrations ↓↓ MDA levels Stabilize blood sugar and prevent blood sugar, liver glycogen, and muscle glycogen levels from reducing in long time exercise on mice Promote the use of fat	[151]
**Clinical Trials**			
Phenolics extracted from *Camellia sinensis*			
	Effects of 7-day supplementation (2000 mg/day) in untrained men subjected to resistance exercise	↓↓ IL-8 levels No effects regarding granulocyte percentage and CD11b adhesion molecule expression levels were observed	[178]
Tea catechins			
	Effects of 3-month supplementation (350 mL of a tea beverage fortified with 540 mg catechins) in elderly women with sarcopenia subjected to 60 min of a comprehensive training program twice a week	↑↑ leg muscle mass and walking speed	[179]
Catechins and theaflavins			
	Effects of 13-week supplementation (2000 mg/day) in active male subjected to eccentric exercise challenge	↓↓ peak torque, whole body and hamstring soreness, serum cortisol, and ↑↑ ferric reducing ability of plasma	[182]
(−)-Epigallocatechin-3-*O*-gallate			
	Effects of 7-day supplementation (135 mg/day) in healthy adults before specific exercise training tests	↑↑ VO_2max_ without affecting maximal cardiac output, and also arterial-venous oxygen difference	[185]
	Effects of 14-day supplementation (1800 mg/day) in skeletal muscle proteolytic gene expression of healthy males after eccentric exercise section	↓↓ muscle ring-finger 1, ubiquitin-protein ligase 3B, and m-calpain expression after exercise	[224]
	Effects of 12-week supplementation (150 mg 2×day) combined with exercise 3 times/week in obese postmenopausal women	↓↓ heart rate and plasma glucose levels	[184]
	Effects of 14-day supplementation (1800 mg/day) in healthy active males after being subjected to a single bout of eccentric muscle contractions daily	↓↓ neutrophils count, CK, 8-isoprostane, cortisol, and TNF-*α* levels No effects regarding cytochrome C, caspase-3 content, caspase-3 enzyme activity, and total DNA were observed	[186]
Anthocyanins juice extracted from apples, plums, blueberries, maqui berries, raspberries, and cranberries			
	Effects of 9-day supplementation (240 mL 2×day) in healthy young men subjected to downhill running	Faster recovery ↓↓ CK levels	[187]
Epicatechin			
	Effects of 8-week supplementation (1 mg/kg/day) in sarcopenic older adults subjected to an intensive training program per week	↑↑ muscle strength and circulatory levels of plasma follistatin and ↓↓ plasma myostatin levels	[183]
Quercetin			
	Effects of 6-week supplementation (300 mg/day) in elite male cyclists before a competition	Better high-intensity cycling performance through the enhancement of power output No differences regarding carbohydrate and fat oxidation were obtained	[160]
	Effects of 2-week supplementation (1000 mg/day) in moderately active young men	Capacity to ↓↓ the severity of muscle weakness caused by eccentric-induced myofibrillar disruption and sarcolemmal action potential propagation impairment ↑↑ isometric strength (+4.7% when compared to baseline)	[161]
	Effects of 2-week supplementation (1000 mg/day) in untrained young male adults	↑↑ mRNA expression of mtDNA and cytochrome c Significant improvement in a 12 min treadmill time trial performance (+15% when compared to placebo)	[225]
	Effects of 1-week supplementation (1000 mg/day) in untrained males	↑↑ VO_2max_ (+3.9%) and ride time to fatigue (+13.2%)	[163]
	Effects of 3-week supplementation (1000 mg/day) before, during, and 2-week after a 3-day of intense exercise (at approximately 57% W_max_) in trained male cyclists	No significant changes in NKCA, PHA-LP, POBA, and sIgA ↓↓ URTI incidence in cyclists after intensified exercise	[133]
	Effects of 8-week supplementation (500 mg/day) in athletic students	↑↑ basal metabolic rate, lean body mass, total body water, and total energy expenditure	[226]
Quercetin conjugated with vitamin C			
	Effects of 8-week daily supplementation with quercetin (500 mg)-vitamin C (250 mg) in men physical education students	Its conjugation with Vitamin C can reduce pro-inflammatory markers and oxidative stress (↓↓ CRP, IL-6, E-selectin, and F2-isoprostane levels)	[227]
	Effects of 8-week daily supplementation with quercetin (500 mg)-vitamin C (200 mg) in male athletes	↓↓ plasma CK levels and body fat percentage	[228]
Quercetin conjugated with epigallocatechin 3-gallate			
	Effects of 2-week daily supplementation with quercetin (1000 mg)-epigallocatechin 3-gallate (120 mg) in trained male and female cyclists	↑↑ granulocyte oxidative burst activity ↓↓ CRP levels No effects regarding peroxisome proliferator-activated receptor γ coactivator *α*, citrate synthase, and cytochrome C were observed	[229]
Resveratrol			
	Effects of 3-month supplementation (100 mg/day) in military firefighters	↓↓ IL-6 and TNF-*α*	[165]
	Effects of 4-day supplementation (480 mg/day) in male athletes subjected to a high-intensity cycling challenge	↓↓ IL-6	[167]
	Effects of 12-week supplementation (500 mg/day) older men and women skeletal muscle	↑↑ mitochondrial density, muscle fatigue resistance, knee extensor muscle peak torque (8%), average peak torque (14%), power (14%), and mean fiber (+45.3%) and total myonuclei (+20%) in muscle fibers	[146]
	Effects of 6-week supplementation (40 mg/day) in male professional basketball players	↓↓ IL-6 and TNF-*α*	[168]
	Effects of 12-week supplementation (250 mg/day) on postexercise endothelial function in estrogen-deficient postmenopausal women	↑↑ basal flow-mediated dilation	[166]
	Effects of supplementation (500 mg/day) on untrained healthy young individuals 3 days prior to isometric ankle dorsiflexion exercises	Attenuate pain perception following exercise-induced muscle damage No effects on force preservation were observed	[230]
	Effects of 8-week supplementation (250 mg/day) in aged man	↑↑ muscle TIMP-1 protein levels	[231]
	Effects of 7-day supplementation (1000 mg/day) in non-athletic men subjected to plyometric exercise	↓↓ muscle damage and inflammation levels, and soreness caused by plyometric-exercise-induced muscle damage	[232]
Resveratrol conjugated with other polyphenolic-rich extracts			
	Effects of 30-day supplementation (60 mg/day) in healthy volunteers	↓↓ IL-6	[233]
Resveratrol combined with piperine			
	Effects of 4-week supplementation (500 mg resveratrol plus 10 mg of piperine) in young adults subjected to forearm wrist flexor exercises	↑↑ forearm skeletal muscle mitochondrial capacity	[172]
Resveratrol combined with quercetin			
	Effects of supplementation with 120 mg resveratrol and 225 mg quercetin for 6 days and 240 mg resveratrol and 450 mg quercetin on day 7 in young adults subjected to forearm wrist flexor	↓↓ F2-isoprostanes levels and exercise-induced lipid peroxidation	[173]
Resveratrol conjugated with carotenoids astaxanthin and *β*-carotene			
	Effects of 10-week supplementation (100 mg/day) in healthy men	↑↑ resistance training-induced strength, metabolic adaptations, and moderated fatigue and oxidative damage	[174]
Ellagitannins from pomegranate extract			
	Effects of 9-day supplementation (500 mL/day) in recreationally active males after a damaging bout of eccentric exercise.	↑↑ strength after 48 and 72 h of exercise No effects regarding serum markers of inflammation and muscle damage were observed	[176]
Hesperetin			
	Effects of 4-week supplementation (500 mg/day) in trained male athletes subjected to cycling time-trial performance	↑↑ absolute power (+5% than placebo) ↓↓ oxygen consumption/power ratio	[177]
Caffeic acid phenethyl ester			
	Effects of 1, 2, and 4 µg/mL exposure in peripheral blood mononuclear cells of competitive cyclists against hyperthermal stress	↓↓ hyperthermia-induced survival inhibition, necrosis, superoxide levels, glutathione depletion	[196]
Curcumin			
	Effects of supplementation (5 g/day) in men 2 days before and to 3 days after eccentric single-leg press exercise	↓↓ in pain during a single-leg squat, gluteal stretch, squat jump, IL-6 levels, and CK activity ↑↑ single-leg jump performance	[234]
	Effects of 8-week supplementation (200 mg/day) in physically active men and women after completion of a downhill running bout	↓↓ peak extension torque values after 1 and 24 h of muscle-damaging exercise	[193]
	Effects of 7-day supplementation (180 mg/day) in healthy men subjected to eccentric exercise	↓↓ IL-8, muscle soreness, and CK activity	[106]
	Effects of 400 mg/day supplementation in 2 days before and 4 days after exercise	↓↓ exercise-induced muscle damage, by lowering CK, TNF-*α*, and IL-8 (-48, -25, and -21% than placebo) No significant differences regarding IL-6 and IL-10 levels and quadriceps muscle soreness were observed	[235]
	Effects of 3-day supplementation (500 mg/day) in non-heat acclimated male and female participants subjected to treadmill runs	↓↓ indicators of cellular energy status SIRT1 and p-AMPK (-47.8 and -48.5% than placebo), and mediators of cellular heat shock response HSP70 protein (-11.0% than placebo) No effects regarding protein expression in peripheral blood mononuclear cells	[236]
	Effects of 3-day supplementation (500 mg/day) in male recreational athletes subjected to 2 h of endurance cycling	Ameliorates psychological stress ↓↓ IL-6 and CK levels	[194]
	Effects of 90 mg supplementation 2 h before and immediately after exercise	Improves antioxidant potential by ↓↓ derivatives of reactive oxygen metabolites	[237]
	Effects of 3-day supplementation (500 mg/day) in non-heat-acclimated humans subjected to treadmill exercises	Improves exertional heat stress responses by ↓↓ I-FABP and IL-1RA, TNF-*α*, and IL-10 levels	[238]
	Effects 4-day supplementation (180 mg/day) in healthy young men after eccentric exercise of the elbow flexors	Ameliorates muscle soreness by ↓↓ CK levels	[192]
	Effects in untrained young men of 150 mg before and 12 hafter being subjected to eccentric exercise	↓↓ maximal voluntary contraction torque and CK levels Faster recovery No effects in IL-6 and TNF-*α* were observed	[239]
	Effects of 150 mg in untrained young men after being subjected to heavy eccentric exercise	↓↓ muscle pain, CK, alanine aminotransferase, and aspartate aminotransferase ↑↑ antioxidant capacity	[191]
	Effects of 3-week supplementation (500 mg/day) in active healthy men subjected to aerobic exercises	↓↓ CK levels and muscle soreness No effects regarding antioxidant capacity, and TNF-*α* and MDA levels were found	[195]
	Effects 400 mg/day supplementation in active healthy men 48 hbefore downhill running test and 24 h after the test	↓↓ muscle pain in the lower limb and IL-8 levels	[240]
	Effects of 200 mg supplementation in moderately trained men before, immediately post, 1-h post, and 24, 48, and 72 h after a downhill running protocol	↓↓ muscle sourness, CK levels Faster recovery	[241]
	Effects on 500 mg supplementation in male recreational athletes followed 2 h of endurance cycling	↓↓ IL-6 and reduce psychological stress	[242]
Curcumin combined with piperine			
	Effects of supplementation (2000 mg of curcumin and 20 mg of piperine×3 times a day) in 48 h before and 48 h of exercise	Improves recovery of the muscle function after the exercise, however, this effect is due to power output loss	[243]
Curcumin combined with *Boswellia serrata*			
	Effects of 12-week supplementation (10 mg of curcumin and 140 mg of *Boswellia*) in master athletes	↓↓ soluble receptor for advanced glycation end-products, advanced glycation end-products, and MDA levels	[244]
Isoflavones			
	Effects of one-year supplementation (75 mg/day) combined with walking 3 times/week in postmenopausal women	↓↓ trunk fat mass	[245]
	Effects of 6-month supplementation (70 mg/day) combined with exercise 3 times/week in obese postmenopausal women	Improvements in body composition parameters (body weight, total and abdominal fat mass, body mass index, appendicular fat-free mass, fat-free mass/fat mass ratio, and sex hormone-binding globulin	[188]
	Effects of 6-month supplementation (70 mg/day) combined with aerobic and resistance training per week in overweight and obese postmenopausal women	↓↓ fat mass and CRP levels	[189]
	Effects of 6-month supplementation (70 mg/day) combined with aerobic and resistance training per week in postmenopausal women	↓↓ fatty liver index and plasma γ-glutamyl-transferase	[190]

VO_2max_, maximal oxygen consumption; FSH, follicle-stimulating hormone; IL, interleukin; TNF-*α*, tumor-necrosis factor-alpha, LDH, lactate dehydrogenase; CK, creatine kinase; MDA, malondialdehyde; mtDNA, mitochondrial DNA; 4-HNE, 4-Hydroxynonenal; 8-OHdG, 8-hydroxy-2′-deoxyguanosin; BCL2, B-cell lymphoma 2; BAX, BCL2 Associated X, Apoptosis Regulator; SkM, skeletal muscle; PDE, phosphodiesterase; CRP, C-reactive protein; NKCA, natural killer cell activity; sIgA, salivary immunoglobulin A; PHA-LP, PHA-stimulated lymphocyte proliferation (PHA-LP); POBA, polymorphonuclear oxidative-burst activity; URTI, upper respiratory tract infection; PGC-1*α*, Peroxisome proliferator-activated receptor-γ coactivator-1*α*; SIRT1, Sirtuin 1; Lxr, liver X receptor; Fxr, Farnesoid X receptor; AST, aspartate aminotransferase; ALT, alanine aminotransferase; ALP, alkaline phosphatase; cGMP, cyclic 3′-5′ guanosine monophosphate; cAMP, cyclic adenosine 3′,5′-monophosphate); CPK, creatine phosphokinase; COX-2, Cyclooxygenase-2; iNOS, inducible nitric oxide synthase; MCP1, monocyte chemoattractant protein-1; ↑↑, increased; ↓↓, decreased.

### 5.2. Phenolic-Rich Sources

Together with individual phenolic compounds, phenolic-rich dietary foods, especially fruits, vegetables, herbs, and their derivatives, have been a subject of many studies and seems to be promising dietary strategies for athletes and sports participants [66]. Indeed, in most cases, the combination of different phenolic subclasses exerts the more notorious health effects [4]. In addition, it is important to consider that phenolic sources are easier to obtain than isolated phenolics because they are widely available on the market [246]. Even so, some care needs to be taken, because in high doses, they can originate adverse effects.

#### 5.2.1. Rodent Models

Among phenolic-rich sources, it has previously been reported that a 10-week supplementation with 0.5% green tea was able to decrease plasma LDH levels and increase muscle lipid *β*-oxidation of mice subjected to swimming workouts. This data is very interesting and can be considered evidence that its supplementation promotes lipids use as an energy source [247,248]. Apples are another functional fruit whose supplementation in sports seems to be very promising. Indeed, a 3-week supplementation (0.05%) in rats subjected to muscle injury induced by stretching contraction showed the capacity to raise glutathione-S-transferase and reduce torque deficits and TBARS levels after eccentric contractions [249]. In young rats subjected to treadmill training, red wine supplementation (25 and 75 mg/kg) showed the ability to improve their endothelial dysfunction and normalize oxidative stress [250]. Moreover, an 8-week supplementation (1 g/kg body weight) of honey by female rats exposed to high-intensity jumping exercises promoted a beneficial effect on bone mass and markers of bone metabolism, as well as a protective effect regarding the disruption of reproductive hormone levels induced by high- and low-intensity jumping exercises [251].

#### 5.2.2. Clinical Trials

Although Amozadeh and colleagues [252] reported that catechin green tea supplementation (33 mg/day) had no effect in obese women when combined with exercise, Sadowska-Krępa et al. [253] reported that a 6-week consumption of green tea (250 mg polyphenols/day) by CrossFit-trained men had a marginal effect on their aerobic capacity. Furthermore, similar supplementation for 4 weeks showed potential to increase total antioxidant capacity and superoxide dismutase activity in erythrocyte and decrease malondialdehyde levels in male sprinters [254] and malondialdehyde and CK levels in sedentary men exposed to strenuous running [255]. On the other hand, a 2-week supplementation (500 mg/day) showed efficacy in reducing fatigue, muscle damage, and oxidative stress in male cyclists, largely due to its ability to lower CK and thiobarbituric acid reactive substances (TBARS) levels [256]. Similar results were obtained by Silva and colleagues [257] who studied the effect in untrained men who underwent certain training sessions. Recently, Magheri et al. [258] also showed that an 8-week supplementation with 500 mg/day reduced IL-6 and CRP levels, body weight, and visceral fat in overweight middle-aged men subjected to endurance training. Moreover, its combination with caffeine (50 mg caffeine and 90 mg green tea) showed thermogenic properties, by increasing energy expenditure and promoting fat oxidation [259]. As for black tea, its intake of 900 mg/day in male students with weight training experiences decreased oxidative stress and IL-6 levels, as well as muscle pain after acute anaerobic exercise [260]. Again, these effects are mainly attributed to the ability of flavan-3-ols to modulate the enzyme catechol-*O*-methyltransferase [180].

In conjunction with red fruits, including grapes, tart cherries, blueberries, and pomegranates have also shown the capacity to enhance performance and promote recovery after various types of strenuous exercises [103,261,262,263,264]. In particular, blackcurrants at a dose of 600 mg/day for one-week improved sprint performance in football players [265], and increased fat oxidation concentrations of non-esterified fatty acids and glycerol in endurance-trained women exposed to prolonged cycling training [266]. Similar results were reported at a dose of 300 mg/day; nevertheless, no effects on heart rate, VO_2max_, and LDH levels were observed [267,268,269]. Additionally, their juice (100 mL/day) showed potential to lower postprandial glycemia, insulinemia, and incretin secretion levels in men and postmenopausal women [270], while 500 mL twice daily increased antioxidant capacity and reduced muscle damage and inflammation, by decreasing CK and IL-6 levels [271]. Regarding blueberry fruits, their extracts (50 µg/mL) have already shown in vitro capacity to suppress induced-muscle damage in rat L6 skeletal muscle cells [272]. In clinical trials, runners who drank 200 mL of blueberry juice twice daily, 2 days before, the day of, and 2 days after a race, showed lower muscle soreness and CRP levels [273]. Its combination with green tea also seems to be an interesting strategy. Indeed, it has already been reported that a 17-day supplementation can significantly reduce the risk of influenza infection after strenuous training [274] and increase ketogenesis in runners during recovery [275]. In addition, the daily supplementation with 250 mL of lemonade made from blueberry, bilberry, cranberry, elderberry, and strawberry fruits and raspberry seeds was reported to be capable to attenuate eccentric-induced impairments in muscle function and improve muscle soreness in sedentary college-aged men [276]. On the other hand, a mixture of blueberries, bananas, and apple juice reduced post-exercise isometric, concentric, and eccentric torque and accelerated recovery and concentric and eccentric strength [277]. These effects are expected and are consistent with other studies. In particular, the intake of bananas is recommended by most athletes and sports practitioners given their richness in carbohydrates and potassium and lower levels of sodium [278]. Besides, a recent study has already reported that these fruits can improve metabolic recovery and reduce COX-2 expression after exercise and the dependence on glycolysis for ATP production [279]. Their combination with pears is also beneficial, being able to improve cycling performance and to reduce iron, blood glucose, insulin, IL-10, cortisol, and total leucocyte count levels, and fatty acid utilization and oxidation after exercise [278].

Regarding apples, physically active men who consumed them together with grapes (500 mg/day) showed better maximal endurance and sense of exertion [280]. Healthy male subjects who underwent an interval swimming test and consumed grapes skins (390 mg × 3 times daily) for 6 weeks showed better swimming times and antioxidant status, as well as lower levels of CK and uric acid [281]. Finally, their combination with pomegranates and green tea at a dosage of 1000 mg/day was effective in increasing, peak maximal and average performance without inducing greater fatigue or higher heart rate, and stabilizing oxidative homeostasis in healthy recreationally active males [262].

Regarding pomegranates, it has been previously shown that the supplementation of pomegranate peels for a period of 3 weeks can attenuate oxidative stress and reduce inflammatory markers in rats exposed to swimming exercises [282]. On the other hand, an 8-week supplementation (815–1350 mg/day) of pomegranate fruits can increase VO_2max_, improve time trial performance in male and female cyclists [283], delay fatigue in treadmill running, and ameliorate vascular diameter and blood flow in active men and women [284]. On the other hand, their juice (500 mL/day) showed the capability to inhibit the type 1 activity of 11 *β*HSD and exert ergogenic effects during intermittent running, promoting beneficial effects on blood flow and reducing oxidative stress in obese and overweight individuals [285], as well as muscle soreness and weakness in resistance-trained men [263]. Participants who underwent 30 min of moderate treadmill training and consumed 500 mL/day for 7 days presented lower levels of malondialdehyde, urinary free cortisol, pre- and post-exercise systolic blood pressure, and lower diastolic blood pressure than placebo [285]. Moreover, 50 mL of their juice over a period of 2 months can accelerate the recovery of strength well-trained rowers [286]. Conjointly, pomegranate juice (250 mL/day) showed the capacity to decrease cortisol and homocysteine levels, and increase the ratio of testosterone/cortisol in elite weightlifters [103], as well as the ability to decrease malondialdehyde levels and increase catalase and glutathione peroxidase activities 48 h after exercise [287]. In addition, a 5-week supplementation of this fruit mixed with mangosteen and black elderberry at a dosage of 500 mg/day showed potential to delay muscle soreness in athletically active men and women, by lowering myoglobin, creatinine, and CK levels [288].

Tart cherry fruits are also marketed as a dietary supplement for athletes. Although the studies of Morehen et al. [289], Abbott et al. [290], and Kupusarevic et al. [291] did not show any improvement in physiological levels with the ingestion of tart cherries, most studies encourage their use by athletes due to their antioxidant and anti-inflammatory properties [105,292,293,294,295,296,297]. Particularly, it was found that a 10-day supplementation of these fruits (480 mg/day) in male resistance athletes can reduce muscle soreness, serum creatinine, total proteins, aspartate aminotransferase, bilirubin, and alanine aminotransferase levels, hence, increasing total antioxidant capacity and improving performance time [295,298]. More recently, Hooper and colleagues [105] found that daily intake of 500 mg/day of cherries can significantly reduce oxidative stress and markers of myocardial damage levels and central fatigue, by decreasing CK and creatine kinase amounts in cardiac muscle. In addition, the daily intake of 30 mL of their juice twice a day in trained men and women showed efficacy in improving recovery and reducing muscle soreness, IL-6, CRP, CK, uric acid, and TBARS concentrations [261,293,296,299,300]. Similar effects were observed at doses of 60, 100, and 240 mL/day [292,297,298,301].

In addition, the supplementation with jabuticaba peel beverage (100 mL/day) 1 h before intense training showed efficacy in attenuating muscle damage and oxidative stress in soccer players, by decreasing urea, alanine aminotransferase, aspartate aminotransferase, and CK levels, and increasing glutathione S-transferase amounts [302].

Citrus fruits can also be considered as a good alternative to synthetic supplements due to their content of flavanones and caffeic acid, whose abilities to improve the sympathetic nervous system have been well-described [114,303]. Indeed, the daily intake of 100 mg of *Citrus aurantium* in combination with 100 mg of caffeine already showed potential to increase plasma epinephrine and norepinephrine levels and decrease blood glucose amounts in physically active men [114]. On the other hand, a 10-day supplementation of lemons (400 mg/day) was effective in accelerating recovery, reducing muscle damage, and increasing glutathione peroxidase activity in physically active men and women [303]. These results are also comparable to those obtained by Kang and coworkers [304]. These authors studied the effect of lychee fruit conjugated with vitamins C and E (200 mg/day) in college students who underwent a series of eccentric knee extensions and concluded that this mixture was also effective in reducing oxidative stress and increasing the anaerobic threshold.

Honey can also be considered another promising energizing substance for athletic activities, not only because of its richness in phenolics but also because of its carbohydrate content [305]. In fact, it had been already reported that the ingestion of 70 g of honey by male road cyclists was effective in counteracting oxidative stress, by increasing superoxide dismutase and catalase activities, and reducing inflammatory cytokines (e.g., IL-1*β*, IL-6, IL-8, and TNF-*α*) [264,306]. It was also mentioned that honey supplementation in healthy volunteer men and female athletes also leads to a reduction in malondialdehyde levels [305,307]. Supplementation with propolis showed comparable effects to honey. Indeed, the daily supplementation (1760 mg) for 9 days showed a remarkable ability to increase total antioxidant capacity and glutathione levels, and decrease IL-6, total oxidative status, malondialdehyde, and oxidative stress index in healthy active subjects [308].

Similar data were observed with dark chocolate supplementation, cocoa, and derivatives. Without surprise, these effects are attributed to the antioxidant effects activity of flavanols and to their ability to modulate glucose uptake [309,310,311,312,313,314,315,316]. Ingestion of 40 mg/day of dark chocolate in male cyclists and elite football players resulted in an increase in free fatty acids during exercise and a decrease of CK and LDH levels after 2 weeks and 1 month of ingestion, respectively [313,317]. A similar dose also showed the capability to promote better time trial performance and improve VO_2max_ in active men [311]. Moreover, their chronic supplementation for 3 months (20 g/day) in healthy and sedentary individuals showed improvements in triglycerides, high-density cholesterol and protein carbonylation levels, mitochondrial efficiency, VO_2max_, and in the activities of LKB1, AMPK, PGC-1*α*, glutathione, and citrate synthase [310]. On the other hand, their acute supplementation (100 g) in male cyclists 2 h before a prolonged exercise showed benefits in terms of plasma glucose and insulin concentrations and antioxidant status [318]. Positive effects have also been observed in soccer players, namely better total antioxidant capacity and lower levels of CK, LDH, carbonyl groups, thiol groups, and malondialdehyde levels [312]. Milk chocolate can also be considered another promising strategy. In fact, its ingestion after strenuous endurance climbing by male athletes showed the ability to reduce muscle soreness when compared to placebo data [319] and prolong the time to complete fatigue in soccer players [315]. On the other hand, two weeks of supplementation (105 g/day) increased the ratio of vitamin E/cholesterol and lowered diastolic blood pressure, malondialdehyde, urate, and LDH levels in young soccer players [316]. In addition, supplementation of 500 mL/day before and 2 h after endurance exercise in male cyclists contributed to faster recovery [320]. A similar dosage was also effective in reducing rapamycin phosphorylation and accelerating time trial and muscle glycogen resynthesis in healthy trained cyclists and triathletes [321]. Finally, it has been recently reported that daily ingestion of 1000 mL in trained male judo athletes subjected to intense training may have beneficial effects on cortisol levels, salivary flow rate, and delayed onset muscle soreness without affecting athletes’ weight [322], improving male cyclists’ recovery after glycogen-lowering training programs [314]. This last data is consistent with those reported by Morgan and colleagues [323]. These researchers studied the effects of 10 days of supplementation (330 mL/day) of cocoa mucilage juice in young and healthy recreationally active men subjected to intensive knee extension 7 days before and 2 days after training and found that the tested group showed a faster recovery. On the other hand, the daily ingestion of 300 mL of cocoa exhibited a remarkable ability to increase the overall antioxidant status of male cyclists at rest and during exercise by lowering uric acid levels [324], while 400 mg/day effectively increased VO_2max_ kinetics in sedentary middle-aged adults subjected to a series of moderate-intensity exercises [325]. In addition, 100 mL of a flavanol enriched cocoa drink can reduce F2-isoprostanes in male volunteers 2 and 4 h after strenuous exercise [326], while one week of supplementation (1765 mg/day) can control lipid peroxidation and oxidative stress during exhaustive exercise in hypoxia and promote improvements in endothelial function and prefrontal oxygenation at rest and during moderate-intensity exercise [327].

Concerning plants, daily supplementation with *Yerba Mate* plant (5 g/day) in well-trained male cyclists 5 days and 1 h before exercise was effective in promoting fat utilization during submaximal exercise and improving their time-trial performance, by increasing power output and VO_2max_ [328]. On the other hand, 40 g of carob powder diluted in 250 mL of water reduced body weight and increased aerobic power rating of perceived exertion in taekwondo athletes [329], while the daily intake of *Ginkgo biloba* leaves (80 mg/day) by healthy and physically active young men increased their endurance performance, VO_2max_ and blood antioxidant capacity [330]. Similar results were reported for *Ecklonia cava* supplementation. This alga demonstrated the ability to improve endurance performance, decrease LDH levels, and increase glucose oxidation in college students subjected to a treadmill running for 30 min [331].

Finally, supplementation with beet juice has also been a target of many studies. Indeed, the daily ingestion of 250 mL has already been shown the ability to reduce muscle pain in healthy male participants exposed to muscle-damaging exercise on day 1 and at the subsequent days [332], and also in soccer players 24 and 48 h after a 100-foot jump [333].

Although further studies are needed to determine appropriate and safe dosing and to rule out external effects, it seems clear that the continued daily supplementation with phenolic-rich sources adds value by increasing well-being and reducing recovery time, muscle soreness, oxidative stress, and proinflammatory markers.

**Table 2 ijms-23-04652-t002:** Main effects of phenolic-rich sources in exercise.

Phenolic-Rich Source	Study Design	Main Outcomes	Reference
Rodent Models			
Apple			
	Preventive effects of 3-week supplementation (0.05%) against lengthening contraction-induced muscle injuries in rats	↓↓ torque deficits after the eccentric contractions, and TBARS protein carbonyl levels ↑↑ glutathione-S-transferase	[249]
Green tea			
	Effects of 10-week supplementation (0.5% green tea) in mice subjected to pool exercises	↑↑ endurance capacity and muscle lipid *β*-oxidation ↓↓ plasma LDH levels Together, both results suggest that lipids were used as an energy source	[247]
	Effects of 10-week supplementation (0.5% green tea) in rats subjected to running exercises	↑↑ metabolic capacity and utilization of fatty acid as a source of energy in skeletal muscle during running exercise	[248]
Honey			
	Effects of 8-week supplementation (1 g/kg body weight) in female rats subjected to 5-day high-intensity jumping exercise	Exert beneficial effects on bone mass and bone metabolism markers Protective effects on disturbance of reproductive hormone levels induced by high and low intensities of jumping exercise	[251]
Pomegranate peel			
	Anti-fatigue effects of 3-week supplementation (25 mg/day) in rats subjected to swimming exercises	↑↑ swimming time and glycogen content without change in liver fat content ↓↓ LDH, CPK, ATP, glycogen content, and MDA levels	[282]
Red wine			
	Effect of 24-week supplementation (25 and 75 mg/kg/day) in young rats subjected to treadmill exercises	↑↑ endothelial dysfunction, normalized oxidative stress and the expression of proteins involved in the formation of nitric oxide, and the angiotensin II pathway	[250]
Clinical Trials			
Blackcurrants fruits			
	Effects of 7-day supplementation (600 mg/day) in trained football players	Benefit repeated sprint performance, by reducing sprint slowing	[265]
	Effects of 7-day supplementation (600 mg/day) in endurance-trained females subjected to prolonged cycling	↑↑ enhances fat oxidation (+27% than placebo) ↑↑ non-esterified fatty acids and glycerol concentrations	[266]
	Effects of 7-day supplementation (300 mg/day) in cyclists	↑↑ fat oxidation (+65% than placebo) and VO_2max_ (+27% than placebo) Improve 16.1 km time trial performance by 2.4%	[334]
	Effects of 7-day supplementation (300 mg/day) in male trained cyclists	↑↑ performance (+0.82% than placebo) No effects regarding heart rate and LDH were observed	[267]
	Effects of 7-day supplementation (300 mg/day) in active male subjected to treadmill running protocol to exhaustion	↑↑ total running distance (+10.6%), distance during sprints increased (+10.8%), and higher LDH values at exhaustion (+15%) No effects regarding heart rate, VO_2max_, and rating of perceived exertion were observed	[268]
	Effects of 7-day supplementation (300 mg/day) in active male with experience in high-intensity intermittent running	Bette and faster time sprint (+50%) No effects regarding LDH, heart rate, time to exhaustion, and vertical jump power were observed	[269]
Blackcurrants juices			
	Effect of 100 mL/day in college students	↓↓ postprandial glycemia, insulinemia and incretin secretion levels	[270]
	Effect of 1-week supplementation (500 mL 2×day) college students subjected to a bout of eccentric knee extensions	↓↓ muscle damage and inflammation, by lowering CK, and IL-6 ↑↑antioxidant capacity	[271]
Blueberries, and bananas fruits, and apple juice			
	Effects of 200 g blueberries, 50 g banana, and 200 mL apple juice supplementation in active females subjected to recreational level resistance and aerobic based exercises	↓↓ isometric, concentric, and eccentric torque followed the exercise Faster rate of recovery, and for concentric and eccentric strength	[277]
Mixed blueberry, bilberry, cranberry, elderberry, raspberry seeds and strawberry lemonade			
	Effects of 250 mL supplementation in sedentary college-aged males before eccentric bout exercises	Mitigate eccentric-induced decrements in muscle function Ameliorates muscle soreness	[276]
Blueberry juice			
	Effects of 2 ×200 mL 5 days before the race, on race day, and 2 days after the race in trained runners	↓↓ delayed onset muscle soreness and CRP	[273]
Lychee fruits combined with vitamins C and E			
	Effect of 1-month supplementation (200 mg/day) in college students subjected to a bout of eccentric knee extensions	↑↑ endurance capacity by increasing anaerobic threshold (+7.4%) ↓↓ oxidative stress	[304]
*Yerba Mate* plant			
	Effect of supplementation (5 g/day) in well-trained male cyclists 5 days and 1 h before experimental trial	↑↑ fat utilization during submaximal exercise and improved time-trial performance, by increasing power output and VO_2max_	[328]
Carob			
	Effects of 6-week supplementation (40 g of carob powder diluted in 250 mL of water) in taekwondo athletes	↑↑ aerobic performance score and rating of perceived exertion ↓↓ body weight No significant effects regarding body fat and muscular volume were found	[329]
Banana			
	Effect of 2-week supplementation of bananas in cyclist men before their 75 Km trial	Improve metabolic recovery and diminish post-exercise inflammation, by ↓↓ COX-2 mRNA expression by THP-1 monocytes and the reliance on glycolysis for ATP production	[279]
Bananas and pears			
	Effect of 2-week supplementation of bananas and pears in cyclist men before their 75 Km trial	Improves 75 Km cycling performance (+5.0% and 3.3% faster regarding banana and pear versus water), by ↑↑ the ferric reducing ability of plasma, blood glucose, and insulin levels ↓↓ fatty acid utilization and oxidation, IL-10, cortisol, and total leucocytes after exercise	[278]
Red grape skin			
	Effects of 6-week supplementation (390 mg × 3 times a day) in healthy male subjects subjected to interval swimming tests	Improves antioxidant status by ↑↑ superoxide dismutase, catalase, glutathione peroxidase, and glutathione reductase activities Faster swimming performance ↓↓ CK and uric acid levels	[281]
Grapes and apples			
	Effect of supplementation (500 mg/day) in physically active men	↑↑ maximal endurance and perceived exertion	[280]
Grapes, pomegranates, and green tea			
	Effect of supplementation (1000 mg/day) in healthy recreationally-active men 1 h before exercise tests	↑↑ total power output (+5%), maximal peak power output (+3.7%), and average power developed (+5%), without inducing more fatigue or greater heart rate Oxidative homeostasis stabilized	[262]
Lemon			
	Effect of 10-day supplementation (400 mg/day) in physically active men and female	Faster and full recovery than placebo ↓↓ muscle damage, exercise-related loss of muscle strength, and movement induced pain ↑↑ glutathione peroxidase activity No effects regarding CK, and IL-6 were observed	[303]
*Citrus aurantium* combined with caffeine			
	Effect of supplementation (100 mg Citrus aurantium + 100 mg caffeine) in physically active men	↓↓ blood glucose levels ↑↑ plasma epinephrine and norepinephrine levels	[114]
Dark chocolate			
	Effects of a 3-month supplementation (20 g/day) in healthy and sedentary individuals	↓↓ triglycerides and protein carbonylation ↑↑ mitochondria efficiency and VO_2max_, high-density cholesterol levels, and LKB1, AMPK, PGC-1*α*, glutathione, and citrate synthase activities	[310]
	Effects of 100 g supplementation in men cyclists 2 h before prolonged exercise	↑↑ pre-exercise antioxidant status and plasma insulin concentration ↓↓ F2-isoprostane levels Better maintenance of plasma glucose concentration No differences regarding stress hormones and IL-6 levels, leukocytosis magnitude and neutrophilia, and changes in neutrophil function were observed	[318]
	Effects of 2-week supplementation (40 g/day) in men cyclists	↓↓ oxidized low-density lipoproteins ↑↑ free fatty acids during exercise No effects regarding circulating insulin, IL-6, IL-10, and IL-1ram glucose, glucagon and cortisol levels, and VO_2max_ were observed	[317]
	Effects of 2-week supplementation (40 g/day) in moderately-trained male participants	↑↑ gas exchange threshold, time trial performance, and VO_2max_	[311]
	Effects of 1-month supplementation (40 g/day) in elite football athletes	↑↑ antioxidant power ↓↓ CK and LDH levels	[313]
	Effects of 25.1 g/day supplementation in soccer players	↓↓ CK, LDH, carbonyl groups, thiol groups, and MDA levels ↑↑ total antioxidant capacity and glutathione peroxidase activity, and in physical performance	[312]
Milk chocolate containing flavanols			
	Effects of 2-week supplementation (105 g/day) in young soccer players	↓↓ diastolic blood pressure, mean blood pressure, plasma cholesterol, LDL, MDA, urate, and LDH ↑↑ vitamin E/cholesterol	[316]
	Effects of 500 mL supplementation in healthy trained cyclists and triathletes immediately before, postexercise, and 2 and 4 h after exercise	Faster time trial and muscle glycogen resynthesis ↓↓ rapamycin phosphorylation	[321]
	Effects of 510 mL supplementation between two training sessions in male and endurance-trained cyclists	↑↑ recovery between both exercise bouts	[335]
	Effects of 240 mL supplementation between two training sessions in soccer players	↑↑ time to fatigue	[315]
	Effects of ≈500 mL in trained male cyclists immediately before and 2 h after endurance exercises	↑↑ recovery after performance	[320]
	Effects of 5-day supplementation (1000 mL/day) in trained male judo athletes subjected to intensive training	↓↓ cortisol, saliva flow rate, and delayed onset muscle soreness ↑↑ salivary testosterone/cortisol ratio No effects regarding weight loss were observed	[322]
	Effects of 5-week supplementation (1000 mL/day) between different training programs in trained male cyclists	Faster recovery after glycogen-lowering exercise	[314]
	Effects of supplementation in male climbers following an exhaustive bout of high-intensity endurance climbing	↓↓ muscle soreness	[319]
Cacao mucilage juice			
	Effect of 10-day supplementation (330 mL/day) in young and healthy recreationally active male subjected to intensive knee extension exercise 7 days before and 2 days after exercise	Faster recovery No effects regarding knee extension maximum voluntary contraction and blood markers were found	[323]
Cocoa			
	Effects of a 3-week supplementation (300 mL/day) in male cyclists	↑↑ total antioxidant capacity in rest and during exercise, by reducing uric acid levels No effects regarding exercise-induced lipid peroxidation, inflammation, nitric oxide production, performance, and recovery were observed	[324]
	Effects of 7-day supplementation (400 mg/day) in sedentary middle-aged adults subjected to a series of ‘step’ moderate- and severe-intensity exercise teat day 7	↑↑ VO_2max_ kinetics during moderate, but not severe-intensity exercise	[325]
Flavanol-rich cocoa drink			
	Effects of a 100 mL supplementation in male volunteers 2 and 4 hafter strenuous physical exercise	↓↓ F2-isoprostanes	[326]
	Effects of 7-day supplementation (1765 mg/day) in well-trained male cyclists	↓↓ lipid peroxidation and oxidative stress during exhaustive exercise in hypoxia Beneficial effects on endothelial function and prefrontal oxygenation at rest and during moderate-intensity exercise	[327]
Tart cherry fruit			
	Effects of a 3 or more days supplementation (500 mg/day) in men subjected to barbell back squat resistance exercises	↓↓ oxidative stress, markers of muscle cardiac damage and central fatigue, by lowering CK and creatine kinase myocardial band content	[105]
	Effects of 10-day supplementation (480 mg/day) in resistance-trained males	↓↓ muscle soreness perception in the vastus medialis and the vastus lateral, serum creatinine and total proteins, AST, bilirubin, and ALT No effects regarding inflammatory and oxidative stress markers were observed	[295]
	Effects of 10-day supplementation (480 mg/day) in male endurance-trained runners and triathletes	Better running performance times ↑↑ total antioxidant capacity ↓↓ deviations from predicted race pace, and creatinine, urea/blood urea nitrogen, total protein, cortisol, and inflammatory markers levels	[298]
Tart cherry juice			
	Effects of 7-day supplementation (50 mL 2×day) in well-trained male cyclists	↓↓ CRP, IL-6 and lipid hydroperoxides	[297]
	Effects of 4-day supplementation (60 mL/day) in male college students subjected to eccentric elbow flexion contractions	↓↓ strength loss and pain	[301]
	Effects of 30 mL 2×day supplementation in well-trained male 7 days before and 48 h after intensive unilateral leg exercise	Improves isometric muscle strength recovery after intensive exercise, by lowering oxidative stress markers ↓↓ CK levels	[292]
	Effects of 8-day supplementation (30 mL 2×day) in trained male cyclists	Maintains muscle function after an exercise stress-induced exclusively through a metabolic challenge ↓↓ IL-6 and CRP	[293]
	Effects of 8-day supplementation (30 mL 2×day) in semi-professional male soccer player	Improves performance indices recovered faster, agility and muscle soreness ↓↓ IL-6 No effects regarding CK and oxidative stress markers were observed	[261]
	Effects of 8-day supplementation (30 mL 2×day) in females subjected to repeated-sprint at day 4	↑↑ recovery of countermovement jump height ↓↓ muscle soreness	[299]
	Effects of 8-day supplementation (30 mL 2×day) in marathon runners	Faster isometric strength recovered ↑↑ total antioxidant capacity ↓↓ IL-6, CRP, uric acid, and TBARS levels No effects regarding protein carbonyls and LDH levels were observed	[300]
	Effects of 8-day supplementation (30 mL 2×day) in male and female sports players	Faster recovery ↓↓ CK levels and muscle soreness	[296]
	Effects of 355 mL 2×day supplementation in healthy runners 7 days before and on the day of the race event	↓↓ pain	[294]
	Effects of 7-day supplementation (240 mL/day supplementation in active individuals subjected to plyometric exercises	↑↑ total antioxidant capacity ↓↓ soreness, CK, LDH, and myeloperoxidase species	[298]
*Camellia sinensis*			
	Effects of 4-week supplementation (320 mg of polyphenols 2×day) in untrained healthy men subjected to strength training	↓↓ plasma lipid hydroperoxides at rest and CK levels	[336]
Green tea			
	Effects of the intake of 3 green tea capsules in healthy young men 24 h before undergoing cycling exercise	↑↑ fat oxidation rates and consequently, ↑↑ fat oxidation to total energy expenditure ↑↑ insulin sensitivity and glucose tolerance	[337]
	Effects of 4-week supplementation (450 mg/day) in male sprinters	Prevents oxidative stress induced by high-intensity cycle sprint test, by decreasing MDA and superoxide radical levels, and probably by inhibiting xanthine oxidase	[254]
	Effects of 6-week supplementation (250 mg/day) in CrossFit individuals	↑↑ ferric reducing ability of plasma ↓↓ TBARS	[253]
	Effects of 3-week consumption (500 mg/day) on whole-body metabolism during cycling exercise in endurance-trained men	↑↑ high-density-lipoprotein cholesterol ↓↓ plasma CK levels No effects regarding fat and energy metabolism, IL-6, CRP, and oxidative stress markers (TBAR and, oxidized low-density-lipoprotein cholesterol) were observed	[338]
	Effect of 15-day supplementation (500 mg/day) in male cyclists	↓↓ muscle damage and oxidative stress, ↓↓ CK and TBARS, and exerts positive effects regarding neuromuscular parameters related to muscle activation and muscle fatigue	[256]
	Effect of 7-day supplementation (2 g of leaves in 200 mL of water, three times per day) in weight-trained men	↓↓ lipid hydroperoxide, CK, AST, reduced glutathione, xanthine oxidase, and uric acid levels before and after exercise ↑↑ the ferric reducing ability of plasma	[339]
	Effects of 4-week supplementation (250 mg/day) in male sprinters	↑↑ total antioxidant capacity and erythrocyte superoxide dismutase activity ↓↓ MDA levels No effects regarding sprint performance were observed	[254]
	Effects of 4-week supplementation (2000 mg/day) in untrained men subjected to resistance training	↑↑ Total antioxidant capacity No effects regarding hinder strength gains were observed	[340]
	Effects of 2-week supplementation (500 mg/day) in untrained men subjected to sessions of exercise to induce delayed onset muscle soreness in the triceps sural muscle group	↓↓ markers of muscle damage after exercise, by lowering CK No effects regarding delayed onset muscle soreness were observed	[257]
	Effects of 4-week supplementation (250 mg/day) in sedentary men subjected to exhaustive run	↑↑ Total antioxidant capacity ↓↓ MDA and CK levels	[255]
	Effects of 20 g of green tea leaves mixed with 600 mL supplementation in well-trained male cyclists during training	Maintenance post-exercise testosterone and lymphocyte concentration ↓↓ neutrophils count	[341]
	Effects of 780 mg/day supplementation in sportive male university gymnastics before training	↓↓ LDH concentration ↑↑ fat oxidation	[180]
	Effects of 8-week supplementation (500 mg/day) in overweight middle-aged men subjected to endurance training	↓↓ IL-6 and CRP levels, and body weight, body mass index, body fat percentage, and visceral fat	[258]
	Effects of 10-week supplementation (572.8 mg/day) in healthy males subjected to a 60 min/day, 3 days/week of ergometer exercises	↓↓ respiratory exchange ratio by increasing the proportion of whole-body fat utilization during exercise	[342]
Green tea combined with caffeine			
	Effects of 24-hcapsule supplementation 3 times/day (50 mg caffeine and 90 mg green tea) in healthy young men	Exhibits thermogenic properties, by increasing energy expenditure and promoting fat oxidation	[259]
Black tea			
	Effects of 4-week supplementation (900 mg/day) in college-age males with weight training experience	↑↑ Recovery ↓↓ oxidative stress levels, IL-6 and delayed onset muscle soreness responses to acute anaerobic intervals	[260]
Propolis			
	Effects of 9-day supplementation (1760 mg/day) in healthy active individuals	↑↑ total antioxidant capacity and glutathione levels ↓↓ IL-6, total oxidant status, MDA, and oxidative stress index No differences regarding IL-10, fat mass, fat-free mass, anaerobic powers, fatigue index, and VO_2max_ were observed	[308]
Honey			
	Effects of 70 g supplementation in healthy nonprofessional male road cyclists before each training session during 8 weeks	↓↓ IL-1*β*, IL-6, IL-8, TNF-*α*, reactive oxygen species and MDA levels ↑↑ superoxide dismutase, catalase, and total antioxidant capacity	[306]
	Effects of 1 g/kg body weight supplementation in healthy volunteer men before exercise during 3 weeks	↓↓ MDA levels	[305]
	Effects of 0.75 and 105 g/kg supplementation in female athletes	Protective effects against lipid peroxidation and oxidative stress by ↓↓ MDA levels	[307]
	Effects of 70 g supplementation 90 min before each training session in male road cyclists during 16 weeks	↓↓ lymphocytes DNA damage, cytokines, peroxidative biomarkers ↑↑ antioxidative biomarkers	[264]
*Ginkgo biloba* leaves			
	Effects of 6-week supplementation (80 mg/day) in healthy and physically active young men	↑↑ endurance performance, VO_2max_ and blood antioxidant capacity, and better neuroprotection through increased exercise-induced production of brain-derived neurotrophic factor, by ↓↓ TBARS and ↑↑ superoxide dismutase, the ferric reducing ability of plasma, and reduced glutathione	[330]
*Ecklonia cava*			
	Effect of 100 mL supplementation in college students 30 min before treadmill tests	Better endurance performance, by ↓↓ LDH levels and ↑↑ glucose oxidation	[331]
Mangosteen, pomegranate, and black elderberry			
	Effects of 5-week supplementation (500 mg/day) in recreationally active men and women	↓↓ delayed onset muscle soreness, by reducing myoglobin, creatinine, and CK levels	[288]
Blueberry and green tea polyphenol-rich soy protein-based product			
	Effects of 17-day supplementation (40 mg/day) in long-distance runners	↑↑ gut-derived phenolic signature and ketogenesis during recovery from 3 days of heavy exertion	[275]
	Effect of 17-day supplementation (12.5 mg/kg/day) against virus infection before an intensive run in long-distance runners	↓↓ the susceptibility to influenza infection following stressful exercise	[274]
Jabuticaba peel beverage			
	Effect of supplementation (100 mL/day) in soccer athletes 1 h before intensive training	Positive effects in attenuating muscle damage and oxidative stress, by lowering urea, ALT, AST, and CK levels, and increasing GST amounts	[302]
Pomegranate			
	Effect of 8-week supplementation (815–1350 mg/day) in male and female cyclists	↑↑ VO_2max_ required during submaximal exercise Ameliorate long time-trials exercises	[283]
	Effect of 3-day supplementation (1000 mg/day and 1300 mg/day) in active male and female	↑↑ vessel diameter and blood flow Delay fatigue during treadmill runs	[284]
Pomegranate juice			
	Effect of 2-day supplementation (250 mL/day) in elite weightlifters	Capacity to attenuate the acute plasma response after exercise, by ↓↓ cortisol and homocysteine levels, and ↑↑ testosterone/cortisol ratio	[103]
	Effect of 250 mL supplementation in elite weightlifters 48 h before training	↑↑ antioxidant responses, by ↓↓ MDA levels (−12.5% than placebo) and ↑↑ catalase and glutathione peroxidase activities (+8.6 and 6.8%, respectively)	[287]
	Effect of 2-week supplementation (250 mL 2×day) in resistance-trained men	Attenuates weakness ↓↓ soreness of the elbow flexor No effects regarding isometric strength and muscle soreness were observed when compared to placebo	[263]
	Effects of 8-day supplementation (650 mg/day and 1300 mg/day) in resistance-trained men	↑↑ strength recovery in leg and arm muscles following eccentric exercise No dose-response effect was observed	[286]
	Effects of 1-week supplementation (500 mL/day) in participants subjected to a 30 min moderate treadmill exercise 2 different occasions	↓↓ MDA, urinary free cortisol, and systolic blood pressure pre-exercise and post- and diastolic blood pressure	[285]
	Effects of 2-month supplementation (50 mL/day) in well-trained rowers	↑↑ plasma antioxidant potential No effects regarding inflammatory markers were observed	[343]
	Effects of 22-day supplementation (200 mL/day) in endurance athletes	Capacity to modulate fat and protein damage by ↑↑ LDH levels and ↓↓ MDA, and carbonyl levels	[344]
	Effects of 1-week supplementation (500 mL/day) in overweight and obese individuals subjected to 30 min of treadmill tests	↓↓ MDA, cortisol, and systolic and diastolic blood pressure before and after exercise	[285]
Beetroot juice			
	Effects 250 mL/day supplementation in healthy male participants subjected to muscle-damaging exercise at day 1 and on the following 3 mornings	↓↓ muscle pain No effects regarding CK, and CPR were observed	[332]
	Effects of 150 mL 2×day 3 days before exercise, on the day trial, and 3 days after exercise in soccer players	↓↓ muscle pain Better performance during the recovery period	[333]
	Effects of 250 mL/day supplementation on the day, 24 h, and 48 h after muscle-damaging exercises in male p	↓↓ countermovement jumps reactive strength index following repeated sprint test, and muscle pain No effects regarding sprint performance or oxidative stress were observed	[345]
	Effects of 250 mL supplementation in soccer players 24, and 48 h after 100-drop jumps	↓↓ muscle soreness No effects regarding maximal isometric voluntary contractions, CK, IL-6, TNF-*α*, and IL-8 were observed	[333]

ALT, alanine aminotransferase; AST, aspartate aminotransferase; CK, creatine kinase; GST, glutathione S-transferase; MAD, malondialdehyde; CK, creatine kinase activity; VO_2max_, maximal oxygen consumption; LKB1, serine/threonine kinase 11, PGC-1*α*, Peroxisome proliferator-activated receptor-γ coactivator-1*α*; TBARS, thiobarbituric acid reactive substance; CRP, C-reactive protein; TBARS, thiobarbituric acid reactive species, LDH, lactate dehydrogenase; VO_2max_, maximal oxygen consumption; ↑↑, increased; ↓↓, decreased.

## 6. Conclusions

In general, the positive and/or negative effects of phenolic supplementation on athletes and sports participants are dependent on the type of training, dose, dietary habits, type of (poly)phenolics consumed, and redox status and training status of the individuals.

For example, in intermittent demanding sports (football, futsal, hockey) where the aerobic and anaerobic component is essential, we recommend the intake of dark chocolate and beetroot derivatives. On the other hand, in sports where aerobic capacity plays a prominent role (cycling, long-distance running), the supplementation with green tea, banana, pears, and red fruits and medicinal plants rich in curcumin is an added value, while in sports where the predominance of anaerobic capacity is important (throwing, jumping, sprints), the ingestion of beetroot juice, green tea, honey and blackcurrant, and tart cherry juices seems to be able to exert positive effects.

Nevertheless, in general, most studies reported that phenolics improve performance during activity, alleviate phycological stress, reduce muscle pain, and restore free radicals to normal levels at rest. However, further studies are needed to investigate the main stable structures, the optimal and safe dosages, and whether supplementation of (poly)phenolics during a strength training program increases or decreases muscle strength and whether they act as pro- or antioxidants. In our opinion, and in accordance with the general directions for research, future works need to consider more detailed measures of dietary control and blinding methods. Additionally, subjects with lower intakes of dietary phenolics will likely respond more favourably to dietary intervention. Finally, there is still a long way to go to understand the full potential of phenolics, including their optimal dosage in exercise and the potential to counteract oxidative stress and avoid uncontrolled inflammatory responses. Even so, it seems to be a promising nutritional strategy to be explored in order to mitigate unwanted sports effects and accelerate recovery.

## Figures and Tables

**Figure 1 ijms-23-04652-f001:**
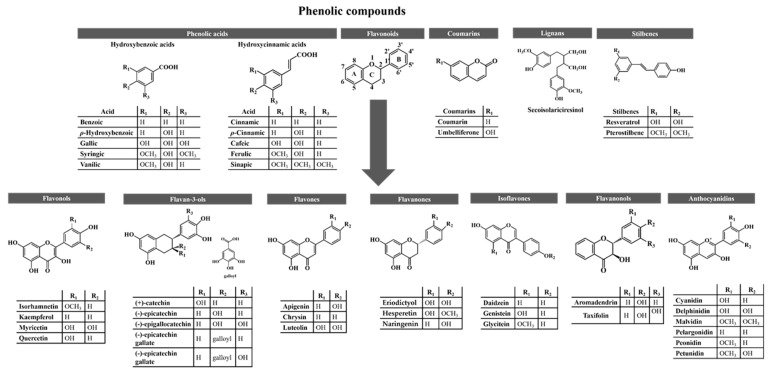
Main phenolic compounds. Non-flavonoids comprise coumarins, phenolic acids, lignans and stilbenes, while flavonoids include flavonols, flavan-3-ols, flavones, isoflavones, flavanonols and anthocyanidins. Among them, flavonoids are the most found in nature.

**Figure 2 ijms-23-04652-f002:**
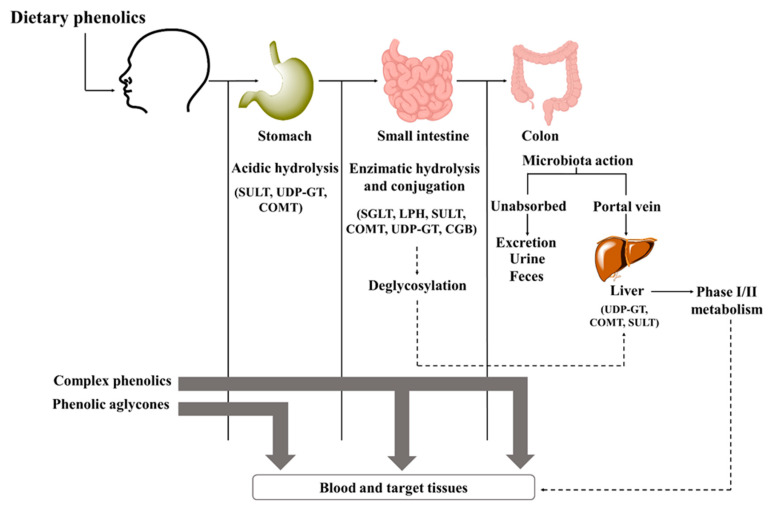
Simplified steps of the metabolic pathways involved in the bioavailability of phenolics in the human body after consumption. CGB—cytosolic *β*-glucosidase; SULT—sulfotransferase; UDP-GT—glucuronosyltransferase; COMT—catechol-O-methyl transferase; SGLT—sodium-dependent glucose cotransporters; LPH—lactase-phlorizin hydrolase.

**Figure 3 ijms-23-04652-f003:**
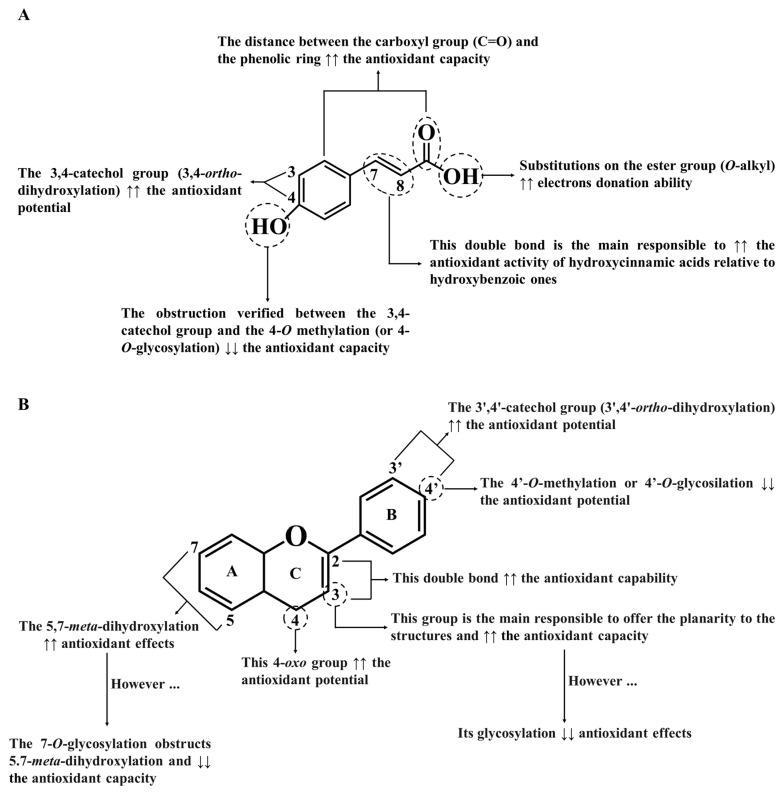
The main structure-activity relationship is responsible to influence the antioxidant capacity of phenolic acids (**A**) and flavonoids (**B**) (adapted from Bento et al. [34]).

**Figure 4 ijms-23-04652-f004:**
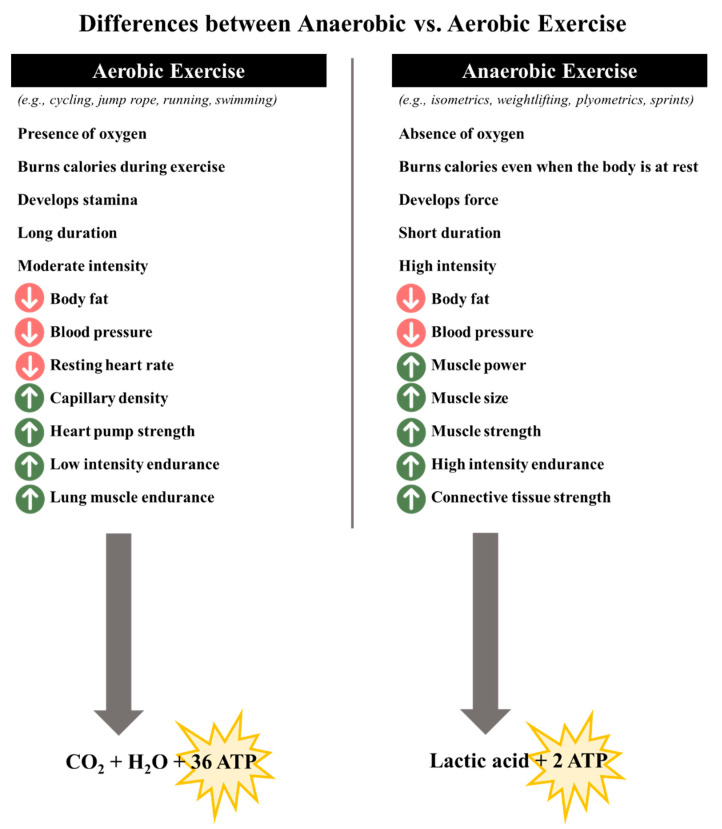
Main differences between aerobic and anaerobic sports.

**Figure 5 ijms-23-04652-f005:**
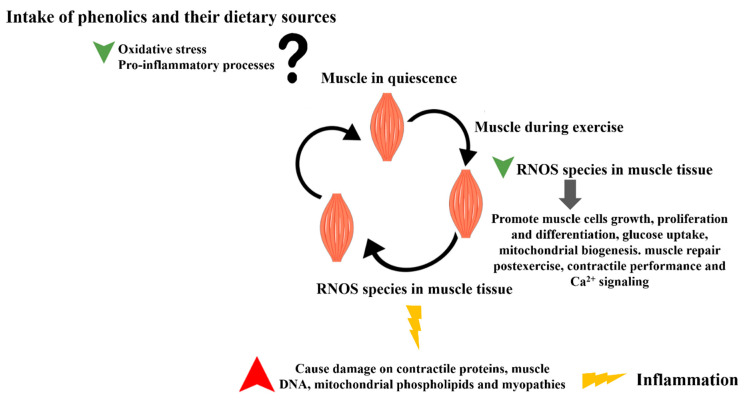
Possible effects of phenolics on athletes and healthcare practitioners’ performance and recovery.

## Data Availability

Not applicable.

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
