# Peer review of "Effects of Functional Phenolics Dietary Supplementation on Athletes’ Performance and Recovery: A Review"

_ijms, 2022, doi:10.3390/ijms23094652_

Round 1
Reviewer 1 Report
The authors in the present review aimed to summarize and discuss the current knowledge on the effects of dietary phenolics on physical performance and recovery in athletes and sports practitioners. Although the work looks huge and efforts of the authors are appreciable, I think that to help the reader understand and follow the logic of the review, different changes will be required. Please find below few comments:
Line 36-37: "This is because, unlike synthetic drugs, it is believed that their use has fewer or no adverse effects"
1. Is unclear to the reviewer the choice of the authors to support their statement with this reference PMID: 30400658
2. Additionally, the sentence is not completely clear.
Example, the content of the sentence itself can be questionable as each bioactive chemical compound should be evaluated singularly, and the claim "natural is always better than synthetic" probably does not belong to science, on the contrary it is easy to fall inside the "gray region".
However, if the authors wanted exactly to point out such concept, "i.e., the common believe is that supplements, nutraceuticals... felt different from synthetic drugs are believed to be more safe, justifying the increase in interest in such aspects...", then the sentence should be reformulated and made more clear.
line 44-45. Again, is unclear to the reviewer the choice of the authors to support their statement with this reference PMID: 30400658. Also, the sentence is too general, as in this way it sounds like there is a consesus about this "potential" added value to sports of phenolic compounds.
As an example, in the study the authors cited few senteces later (line 49) PMID: 33356471, the results of the systematic review with meta-analysis revealed: 1. discordant results concerning training adaptations, 2. resveratrol induced non-functional adaptations or impaired exercise-induced adaptations.
Please consider to not overclaim and to be more precise.
lines 46-48 Why the authors used PMID: 33356471 as a citation for this sentence, when the article discuss about the impact of (poly)phenolic supplements on training adaptations, and only secondarily/partially mentioned aspects about exercise-induced damages and its possible consequences. Please consider to rewrite or to chose a proper reference for your statement that anyway could be improved.
lines 50-51 Again, although the study mentioned by the authors refers to the impact of ROS and the role of montmorency cherries (rich in polyphenols) on exercise performance, it is unclear why the authors mentioned 1. the fact that in pre-season of most sports more than one training session per-day could be carried out and 2. where is demonstrated that just the increase on training frequency can determine an increase in RNOS and pro-inflammatory markers? Rather the characteristics of the training may better lead to greater or lower exercise-induce muscle damage and inflammatory biomarkers rise.
lines 56-58 To support this sentence tha authors cited:
1. a case study (5)
2. two studies on montmorency cherries
1. one systematic review with meta analysis that revealed discordant or at least non-univocal results regarding polyphenols supplementation and exercise adaptations.
Please eliminate or rewrite without overclaims.
line 61 Again, the authors discuss about precise supplementation strategies and its effects but cited a review. In addition the conclusion of the review have nothing or almost nothing to do with athletic performance enhancement. Please cite the right source or cancel/rewrite the sentence.
lines 70-72. Again the reference does not make sense according to the sentence.
Kang, J.Y.; Park, S.K.; Guo, T.J.; Ha, J.S.; Lee, D.S.; Kim, J.M.; Lee, U.; Kim, D.O.; Heo, H.J. Reversal of trimethyltin-induced learning and 843 memory deficits by 3,5-dicaffeoylquinic acid. Oxid. Med. Cell. Longev. 2016, 2016, doi:10.1155/2016/6981595
After spotting strange associations with references in the text I suggest the authors to double check their reference since is possible that something went wrong during paper formatting or resubmitting it. In this case I am sorry for the comments above but it was hard to read till this point in this way.
Differently if is not a formatting mistake, I suggest to check well again the text and reformulate sentences and references because in this way is hard to read, despite it looks a huge work by the authors in collecting and reviewing all these information.
Author Response
Response to Reviewers’ Comments (Manuscript ijms-1675548)
Ms. Ref. No. ijms-1675548
Title: Effects of functional phenolics dietary supplementation on athletes’ performance and recovery: a review
Reviewer 1
The authors in the present review aimed to summarize and discuss the current knowledge on the effects of dietary phenolics on physical performance and recovery in athletes and sports practitioners. Although the work looks huge and efforts of the authors are appreciable, I think that to help the reader understand and follow the logic of the review, different changes will be required. Please find below few comments:
Authors’ response: First of all, we would like to thank your kind comments and compliments made on our manuscript. Following the comments received, the changes made by us are highlighted in the revised version, in accordance with the request. In addition, English was all revised. Even so, after each Reviewer’s comment, we indicated the main changes introduced and the corresponding lines. Thank you.
Line 36-37: "This is because, unlike synthetic drugs, it is believed that their use has fewer or no adverse effects"
- Is unclear to the reviewer the choice of the authors to support their statement with this reference PMID: 30400658
- Additionally, the sentence is not completely clear.
Example, the content of the sentence itself can be questionable as each bioactive chemical compound should be evaluated singularly, and the claim "natural is always better than synthetic" probably does not belong to science, on the contrary it is easy to fall inside the "gray region".
However, if the authors wanted exactly to point out such concept, "i.e., the common believe is that supplements, nutraceuticals... felt different from synthetic drugs are believed to be more safe, justifying the increase in interest in such aspects...", then the sentence should be reformulated and made more clear.
Authors’ response: Thank you so much for your appointment. You have total reason, and therefore, the sentence was all re-written (please see now lines 36 to 38 of the revised version and also the new references added.).
line 44-45. Again, is unclear to the reviewer the choice of the authors to support their statement with this reference PMID: 30400658. Also, the sentence is too general, as in this way it sounds like there is a consesus about this "potential" added value to sports of phenolic compounds.
Authors’ response: Thank you so much for your appointment. The sentence was all revised and new references were added (please see now lines 42 to 53 of the revised version and also the new references added.).
As an example, in the study the authors cited few senteces later (line 49) PMID: 33356471, the results of the systematic review with meta-analysis revealed: 1. discordant results concerning training adaptations, 2. resveratrol induced non-functional adaptations or impaired exercise-induced adaptations. Please consider to not overclaim and to be more precise.
Authors’ response: Thank you so much for your appointment. Although the chosen reference was only to justify the sentence, we changed the reference to be more in accordance with the mentioned (please see now the revised version).
lines 46-48 Why the authors used PMID: 33356471 as a citation for this sentence, when the article discuss about the impact of (poly)phenolic supplements on training adaptations, and only secondarily/partially mentioned aspects about exercise-induced damages and its possible consequences. Please consider to rewrite or to chose a proper reference for your statement that anyway could be improved.
Authors’ response: Thank you so much for your appointment. In accordance with the suggested, we decided to change the reference to be more in accordance with the mentioned (please see now the revised version).
lines 50-51 Again, although the study mentioned by the authors refers to the impact of ROS and the role of montmorency cherries (rich in polyphenols) on exercise performance, it is unclear why the authors mentioned 1. the fact that in pre-season of most sports more than one training session per-day could be carried out and 2. where is demonstrated that just the increase on training frequency can determine an increase in RNOS and pro-inflammatory markers? Rather the characteristics of the training may better lead to greater or lower exercise-induce muscle damage and inflammatory biomarkers rise.
Authors’ response: Thank you so much for your appointment. In accordance with the suggested, we decided to change the reference to be more in accordance with the mentioned (please see now the revised version).
lines 56-58 To support this sentence tha authors cited: 1. a case study (5) 2. two studies on montmorency cherries 1. one systematic review with meta analysis that revealed discordant or at least non-univocal results regarding polyphenols supplementation and exercise adaptations. Please eliminate or rewrite without overclaims.
Authors’ response: Thank you so much for your appointment. In accordance with the suggested, we decided to change the reference to be more in accordance with the mentioned (please see now the revised version).
line 61 Again, the authors discuss about precise supplementation strategies and its effects but cited a review. In addition the conclusion of the review have nothing or almost nothing to do with athletic performance enhancement. Please cite the right source or cancel/rewrite the sentence.
Authors’ response: Thank you so much for your appointment. In accordance with the mentioned, we decided to change the reference to be more in accordance with the mentioned (please see now the revised version).
lines 70-72. Again the reference does not make sense according to the sentence.
Kang, J.Y.; Park, S.K.; Guo, T.J.; Ha, J.S.; Lee, D.S.; Kim, J.M.; Lee, U.; Kim, D.O.; Heo, H.J. Reversal of trimethyltin-induced learning and 843 memory deficits by 3,5-dicaffeoylquinic acid. Oxid. Med. Cell. Longev. 2016, 2016, doi:10.1155/2016/6981595
Authors’ response: Thank you so much for your appointment. In accordance with the mentioned, we decided to change the reference to be more in accordance with the mentioned (please see now the revised version).
After spotting strange associations with references in the text I suggest the authors to double check their reference since is possible that something went wrong during paper formatting or resubmitting it. In this case I am sorry for the comments above but it was hard to read till this point in this way.
Authors’ response: Thank you so much. The article was all revised.
Differently if is not a formatting mistake, I suggest to check well again the text and reformulate sentences and references because in this way is hard to read, despite it looks a huge work by the authors in collecting and reviewing all these information.
Authors’ response: Thank you so much. As suggested, the article was all revised and some sentences were clarified in order to facilitate their read.

Reviewer 2 Report
The aim of this narrative review is to provide a detailed review of the effects of the major dietary sources of phenolics on the performance and recovery of athletes and sports practitioners. This overview is divided into three parts. The first two parts described the major dietary phenolic sources, their bioavailability, and the relationship between their structure and health benefits. Finally, the third part outlined the latest findings and discussion on the main effects of phenolics in athletes and sports practitioners.
Despite that the review was well written, however, I don't know exactly:
- The difference between this narrative review and the previously published reviews
- Why the authors did not perform a systematic review of the literature (with a high level of scientific evidence) instead of a narrative review? In general, narrative reviews are subjective.
- The authors should indicate the databases used in their articles' search.
- What are the inclusion and exclusion criteria?
I am sure that the authors have done huge work, but further focus on the rationale of the current review and its transparency by indicating databases and eligibility criteria is needed. The authors may have "forgotten" to include some interesting articles in this review.
Author Response
Response to Reviewers’ Comments (Manuscript ijms-1675548)
Ms. Ref. No. ijms-1675548
Title: Effects of functional phenolics dietary supplementation on athletes’ performance and recovery: a review
Reviewer 2
The aim of this narrative review is to provide a detailed review of the effects of the major dietary sources of phenolics on the performance and recovery of athletes and sports practitioners. This overview is divided into three parts. The first two parts described the major dietary phenolic sources, their bioavailability, and the relationship between their structure and health benefits. Finally, the third part outlined the latest findings and discussion on the main effects of phenolics in athletes and sports practitioners.
Authors’ response: First of all, we would like to thank your kind comments and compliments made on our manuscript. Following the comments received, the changes made by us are highlighted in the revised version, in accordance with the request. In addition, English was all revised. Even so, after each Reviewer’s comment, we indicated the main changes introduced and the corresponding lines. Thank you.
Despite that the review was well written, however, I don't know exactly:
- The difference between this narrative review and the previously published reviews
Authors’ response: Thank you so much for your note. Although there were other review studies, this is the first one that distinguishes between aerobic and/or anaerobic exercises and the requirements of each one, and also detailed explains why phenolics can be a promising strategy.
- Why the authors did not perform a systematic review of the literature (with a high level of scientific evidence) instead of a narrative review? In general, narrative reviews are subjective.
Authors’ response: Thank you so much for your note. In fact, we realized a systematic review. This information was added (please see now lines 78 to 99 of the revised version).
- The authors should indicate the databases used in their articles' search.
Authors’ response: Thank you so much for your note. This was added (please see now lines 78 to 99 of the revised version).
- What are the inclusion and exclusion criteria?
Authors’ response: Thank you so much for your note. This information was added (please see now lines 78 to 99 of the revised version).
I am sure that the authors have done huge work, but further focus on the rationale of the current review and its transparency by indicating databases and eligibility criteria is needed. The authors may have "forgotten" to include some interesting articles in this review.
Authors’ response: Thank you so much, the database used and eligibility criteria were added, as well as other relevant articles (please see now lines 78 to 99 of the revised version, and also the new references topic).

Round 2
Reviewer 1 Report
The authors in the present review aimed to summarize and discuss the current knowledge on the effects of dietary phenolics on physical performance and recovery in athletes and sports practitioners. After the major revision from the authors the work now is more easy to read and the referenced works make more sense in supporting their statements.
Few additional suggestions:
Line 50-52: as mentioend in the previous revision, not only eccentric or explosive training or exercises generate muscle damage but in the context of professional athletes the factors that lead to muscle damage can be enormous. Therefore, although your statement is correct, could be better to place in this case a more general sentence or if you want to be specific you should cover all aspects (or at least the main) leading to muscle damage in athletes. This is a minor consideration but can be important since your work is directed to a wide spectrum of population (athletes or who work with them).
line 57-63: This sentence sounds to me still as an overclaim not completely supported by evidence, consider to make it "softer".
Line 75-95: This new section makes sense and it helps to understand the authors' approach to the review, however, it should be improved at least in its form (the contents can be fine).
Sections 3 and 4 are well written and comprehensive
Section 5
Line 366-369: again, the same as mentioned for above for line 50-52. The factors leading to muscle damage, soreness or injuries are many and for sure not limited to exposition to eccentric or explosive movements, trainings or exercises. Especially when you refere to the "global" athletic population. Oxidative stress maybe related just to the intensity of exercise or experience of the individual, indipendently from the exercise task (e.g., https://www.frontiersin.org/articles/10.3389/fphys.2020.610112/full; https://pubmed.ncbi.nlm.nih.gov/20308979/) Also, just cycling competitions (without any specific training as the authors mentioned resisted cycling) induces a rise in oxidative stress markers.
What about, in this section, the possible positive roles of pro-inflammatory cytokines or similar, in order to elicit the adaptive response leading to improvements in morphological aspects of the muscle tendon unit and/or cardiorespiratory system structures and consequently improving physical performance?
Structural damage is not always bad for athletes, on the contrary is a goal to induce damages and then rocover, building the bases for improvements.
Is clear what the authors pointed out: an excess of oxidative stress may limit performance at different levels (health, training adaptations, recovery, and ultimately physical performance globally). However, in absence of precise tools to accurately determine the limit between functional and non-functional stress for the athlete, what are the risk of supplementing when not necessary in terms of training adaptations, and how to find the right moment to prescribe a supplementation for the athlete?
Table 1 and Section 5. I suggest the authors to discuss and represent separately studies on humans or animal models.
In general section 5 should be revised and the literature discussed more critically thinking deeply about all aspects sorrounding athletes' performance and not just reporting the positive effects (or mostly) of one or the other supplement.
The same for the conclusion. The authors tend to overclaim the effects of phenolic supplements.
In general the work improved consistently, however it can still become better with few changes especially in the sections mentioned.
Author Response
Response to Reviewers’ Comments (Manuscript ijms-1675548)
Ms. Ref. No. ijms-1675548
Title: Effects of functional phenolics dietary supplementation on athletes’ performance and recovery: a review
Reviewer 1
The authors in the present review aimed to summarize and discuss the current knowledge on the effects of dietary phenolics on physical performance and recovery in athletes and sports practitioners. After the major revision from the authors the work now is more easy to read and the referenced works make more sense in supporting their statements. Few additional suggestions:
Authors’ response: First of all, we would like to thank your kind comments and compliments made on our manuscript. Following the comments received, the changes made by us are highlighted in the revised version, in accordance with the request. In addition, English was all revised. Even so, after each Reviewer’s comment, we indicated the main changes introduced and the corresponding lines. Thank you.
Line 50-52: as mentioend in the previous revision, not only eccentric or explosive training or exercises generate muscle damage but in the context of professional athletes the factors that lead to muscle damage can be enormous. Therefore, although your statement is correct, could be better to place in this case a more general sentence or if you want to be specific you should cover all aspects (or at least the main) leading to muscle damage in athletes. This is a minor consideration but can be important since your work is directed to a wide spectrum of population (athletes or who work with them).
Authors’ response: Thank you so much for your commentary. As suggested, we clarified this sentence (please see now lines 48 to 56 of the revised version).
line 57-63: This sentence sounds to me still as an overclaim not completely supported by evidence, consider to make it "softer".
Authors’ response: Thank you so much for your appointment. As recommended, the sentence was revised in order to make it “softer” (please see now lines 55 to 64 of the revised version).
Line 75-95: This new section makes sense and it helps to understand the authors' approach to the review, however, it should be improved at least in its form (the contents can be fine).
Authors’ response: Thank you so much for your note. The section was revised to stay clearer (please see now section 2, lines 81 to 101 of the revised version).
Sections 3 and 4 are well written and comprehensive
Authors’ response: Thank you so much for your opinion.
Section 5
Line 366-369: again, the same as mentioned for above for line 50-52. The factors leading to muscle damage, soreness or injuries are many and for sure not limited to exposition to eccentric or explosive movements, trainings or exercises. Especially when you refere to the "global" athletic population. Oxidative stress maybe related just to the intensity of exercise or experience of the individual, indipendently from the exercise task (e.g., https://www.frontiersin.org/articles/10.3389/fphys.2020.610112/full; https://pubmed.ncbi.nlm.nih.gov/20308979/) Also, just cycling competitions (without any specific training as the authors mentioned resisted cycling) induces a rise in oxidative stress markers.
Authors’ response: Thank you so much for your note. As recommended, the sentence was re-written (please see now lines 360 to 370 of the revised version).
What about, in this section, the possible positive roles of pro-inflammatory cytokines or similar, in order to elicit the adaptive response leading to improvements in morphological aspects of the muscle tendon unit and/or cardiorespiratory system structures and consequently improving physical performance?
Authors’ response: Thank you so much for your suggestion. In fact, the positive role of inflammation was already mentioned in the article. Even so, we added more information (please see now lines 373 to 392 of the revised version). Additionally, and in line with your commentary, it is important to note that “proinflammatory cytokines seem to play a key role in muscle atrophy by regulating the pathways involved in this condition. As such, they may cause severe decrease in muscle strength and power, as well as impairment in cardiorespiratory fitness” (doi:10.1155/2016/395795). Therefore, until now, it is well-described the positive roles of inflammation only in the early stages to create strength, promote muscle repair and regeneration, re-adaptation redox signaling pathways for the normal state, optimize vasodilation, mitochondrial reparation, and hyperemia during exercise), however, when there is verified an imbalance (e.g., muscular catabolism, fiber damage, impairment of neuromuscular functions, training adaptations, among others), immunosuppressive events, extreme fatigue, and excessive inflammatory reactions may occur, it can become harmful for human health, and limit the performance in competitions. These informations were referred in the article.
Structural damage is not always bad for athletes, on the contrary is a goal to induce damages and then recover, building the bases for improvements.
Authors’ response: Thank you so much for this note and you have total reason. The same was mentioned (please see now lines 360 to 365 of the revised version).
Is clear what the authors pointed out: an excess of oxidative stress may limit performance at different levels (health, training adaptations, recovery, and ultimately physical performance globally). However, in absence of precise tools to accurately determine the limit between functional and non-functional stress for the athlete, what are the risk of supplementing when not necessary in terms of training adaptations, and how to find the right moment to prescribe a supplementation for the athlete?
Authors’ response: Thank you so much for your note. The same information was added (please see now lines 412 to 415, and 446 to 466 of the revised version). Additionally, we already mentioned in conclusion section that deeper studies are needed to full reveal the full potential of phenolics, including their optimal dosage in exercise and the potential to counteract oxidative stress and avoid uncontrolled inflammatory responses.
Table 1 and Section 5. I suggest the authors to discuss and represent separately studies on humans or animal models.
Authors’ response: Thank you so much for your suggestion. As recommended, we divided the table and section based on animal models and humans (please see now tables 1 and 2, and section 5 of the revised version).
In general section 5 should be revised and the literature discussed more critically thinking deeply about all aspects sorrounding athletes' performance and not just reporting the positive effects (or mostly) of one or the other supplement.
Authors’ response: Thank you much for your suggestion. As recommended, section 5 was deeper discussed (please see now section 5 of the revised version).
The same for the conclusion. The authors tend to overclaim the effects of phenolic supplements.
Authors’ response: Thank you so much for your appointment. As suggested, the conclusion was ameliorated (please see this section in the revised version).
In general the work improved consistently, however, it can still become better with few changes especially in the sections mentioned.
Authors’ response: Thank you so much for your suggestions, which in fact, enriched the present article.

Reviewer 2 Report
The authors did not perform a systematic review. Indeed, the PRISMA figure is lacking, which makes this study not scientifically sound. I recommend the rejection of the manuscript.
Author Response
Response to Reviewers’ Comments (Manuscript ijms-1675548)
Ms. Ref. No. ijms-1675548
Title: Effects of functional phenolics dietary supplementation on athletes’ performance and recovery: a review
Reviewer 2
The authors did not perform a systematic review. Indeed, the PRISMA figure is lacking, which makes this study not scientifically sound. I recommend the rejection of the manuscript.
Authors’ response: Thank you so much for your commentary. However, the authors have spoken with each other, and, with lots of sorry, we thought that there is some misunderstanding. Firstly, in section 2, the authors explained how they did the collection of data. As mentioned in section 2, the authors, through specific databases and using specific words, combined or not, searched and downloaded the most significant articles to perform this detailed review (not systematic nor overview nor meta-analysis), but yes detailed, sorry for the misunderstanding). This type of review and search is in accordance with other ones published with high relevance, and therefore, it is not only systematic reviews that can be published. In fact, most review articles use this type of search data. In addition, as we both know, systematic reviews are more detailed and indicated for medical issues “A systematic review is a summary of the medical literature that uses explicit and reproducible methods to systematically search, critically appraise, and synthesize on a specific issue. It synthesizes the results of multiple primary studies related to each other by using strategies that reduce biases and random errors. [7] To this end, systematic reviews may or may not include a statistical synthesis called meta-analysis, depending on whether the studies are similar enough so that combining their results is meaningful. [8] Systematic reviews are often called overviews.” (doi: 10.4103/2249-4863.109934), which is not the case. In fact, as mentioned in the article, the main focus of this is to mention the most relevant studies published until now, discuss, i.e., review, the role, and biological potential of phenolics and phenolic-rich sources in athletes’ performance and recovery. Additionally, most review articles did not even indicate how they collected the data (which is not the case, since we mentioned it in topic 2 to complete the article). Given that, please see:
Golovinskaia, O.; Wang, C.K. Review of functional and pharmacological activities of berries. Molecules 2021, 26, doi:10.3390/molecules26133904.
Hughes, K.M.; Price, D.; Torriero, A.A.J.; Symonds, M.R.E.; Suphioglu, C. Impact of Fungal Spores on Asthma Prevalence and Hospitalization. Int. J. Mol. Sci. 2022, 23, doi:10.3390/ijms23084313.
Truzzi, F.; Tibaldi, C.; Zhang, Y.; Dinelli, G.; D′Amen, E. An overview on dietary polyphenols and their biopharmaceutical classification system (BCS). Int. J. Mol. Sci. 2021, 22, doi:10.3390/ijms22115514.
Magagula, L.P.; Masemola, C.M.; Ballim, M.A.; Tetana, Z.N.; Moloto, N.; Linganiso, E.C. Lignocellulosic biomass waste-derived cellulose nanocrystals and carbon nanomaterials: A review. Int. J. Mol. Sci. 2022, 23, 4310, doi: 10.3390/ijms23084310.
Sultana, T.; Jan, U.; Lee, J.I. Double repositioning: Veterinary antiparasitic to human anticancer. Int. J. Mol. Sci. 2022, 23, 4315, doi: 10.3390/ijms23084315.
Kamperman, R.G.; Kooi, A.J. Van Der; Visser, M.; Aronica, E.; Raaphorst, J. Pathophysiological mechanisms and treatment of dermatomyositis and immune mediated necrotizing myopathies: A focused review. Int. J. Mol. Sci. 2022, 23, doi:10.3390/ijms23084301
Lange, N.F.; Graf, V.; Caussy, C.; Dufour, J.-F. PPAR-Targeted therapies in the treatment of non-alcoholic fatty liver disease in diabetic patients. Int. J. Mol. Sci. 2022, 23, doi:10.3390/ijms23084305.
Takasawa, S. CD-38 Cyclic ADP-Ribose signal system in physiology, biochemistry and pathophysiology. Int. J. Mol. Sci. 2022, 23, doi:10.3390/ijms23084306.
Swolana, D.; Wojtyczka, R.D. Activity of silver nanoparticles against Staphylococcus spp. Int. J. Mol. Sci. 2022, 23, doi:10.3390/ijms23084298.
Huang, J.-K.; Lee, H.-C. Emerging Evidence of pathological roles of very-low-density lipoprotein (VLDL). Int. J. Mol. Sci. 2022, 23, doi:10.3390/ijms23084300.
Jang, J.Y.; Sung, B.; Kim, N.D. Role of induced programmed cell death in the chemopreventive potential of apigenin. Int. J. Mol. Sci. 2022, 23, doi:10.3390/ijms23073757.
Restivo, I.; Attanzio, A.; Tesoriere, L.; Allegra, M.; Garcia-Llatas, G.; Cilla, A. Anti-eryptotic activity of food-derived phytochemicals and natural compounds. Int. J. Mol. Sci. 2022, 23, doi:10.3390/ijms23063019.
Tosif, M.M.; Najda, A.; Bains, A.; Krishna, T.C.; Chawla, P.; Dyduch-Siemińska, M.; Klepacka, J.; Kaushik, R. A comprehensive review on the interaction of milk protein concentrates with plant-based polyphenolics. Int. J. Mol. Sci. 2021, 22, doi:10.3390/ijms222413548.
Stiller, A.; Garrison, K.; Gurdyumov, K.; Kenner, J.; Yasmin, F.; Yates, P.; Song, B.H. From fighting critters to saving lives: Polyphenols in plant defense and human health. Int. J. Mol. Sci. 2021, 22, doi:10.3390/ijms22168995.
Truong, V.L.; Jeong, W.S. Cellular defensive mechanisms of tea polyphenols: Structure-activity relationship. Int. J. Mol. Sci. 2021, 22, doi:10.3390/ijms22179109.
Ávila-Gálvez, M.Á.; Giménez-Bastida, J.A.; Espín, J.C.; González-Sarrías, A. Dietary phenolics against breast cancer. A Critical Evidence-Based Review and Future Perspectives. Int. J. Mol. Sci. 2020, 21, doi:10.3390/ijms21165718.
Gómez-Maqueo, A.; Escobedo-Avellaneda, Z.; Welti-Chanes, J. Phenolic compounds in Mesoamerican fruits—Characterization, health potential and processing with innovative technologies. Int. J. Mol. Sci. 2020, 21, doi:10.3390/ijms21218357.
Shahidi, F.; Yeo, J. Bioactivities of phenolics by focusing on suppression of chronic diseases: A review. Int. J. Mol. Sci. 2018, 19, 1573, doi: 10.3390/ijms19061573.
Nirumand, M.C.; Hajialyani, M.; Rahimi, R.; Farzaei, M.H.; Zingue, S.; Nabavi, S.M.; Bishayee, A. Dietary plants for the prevention and management of kidney stones: Preclinical and clinical evidence and molecular mechanisms. Int. J. Mol. Sci. 2018, 19, doi:10.3390/ijms19030765.
Skrovankova, S.; Sumczynski, D.; Mlcek, J.; Jurikova, T.; Sochor, J. Bioactive compounds and antioxidant activity in different types of berries. Int. J. Mol. Sci. 2015, 16, 24673–24706, doi:10.3390/ijms161024673.
de Pascual-Teresa, S.; Moreno, D.A.; García-Viguera, C. Flavanols and anthocyanins in cardiovascular health: A review of current evidence. Int. J. Mol. Sci. 2010, 11, 1679–1703, doi:10.3390/ijms11041679.
Bounegru, A.V.; Apetrei, C. Laccase and tyrosinase biosensors used in the determination of hydroxycinnamic acids. Int. J. Mol. Sci. 2021, 22, doi:10.3390/ijms22094811
Gonçalves, A.C.; Flores-Félix, J.D.; Coutinho, P.; Alves, G.; Silva, L.R. Zimbro (Juniperus communis L.) as a promising source of bioactive compounds and biomedical activities: A review on recent trends. Int. J. Mol. Sci. 2022, 23, doi:10.3390/ijms23063197.
Singla, R.K.; Dubey, A.K.; Garg, A.; Sharma, R.K.; Fiorino, M.; Ameen, S.M.; Haddad, M.A.; Al-Hiary, M. Natural polyphenols: Chemical classification, definition of classes, subcategories, and structures. J. AOAC Int. 2019, 102, 1397–1400, doi:10.5740/jaoacint.19-0133.
Tungmunnithum, D.; Thongboonyou, A.; Pholboon, A.; Yangsabai, A. Flavonoids and other phenolic compounds from medicinal plants for pharmaceutical and medical aspects: An overview. Medicines 2018, 5, 93, doi:10.3390/medicines5030093.
Raina, R.; Verma, P.K.; Peshin, R.; Kour, H. Potential of Juniperus communis L as a nutraceutical in human and veterinary medicine. Heliyon 2019, 5, e02376, doi:10.1016/j.heliyon.2019.e02376.
Tanase, C.; Cosarcă, S.; Muntean, D.L. A critical review of phenolic compounds extracted from the bark of woody vascular plants and their potential biological activity. Molecules 2019, 24, doi:10.3390/molecules24061182.
Zhang, Y.; Cai, P.; Cheng, G.; Zhang, Y. A brief review of phenolic compounds identified from plants: Their extraction, analysis, and biological activity. Nat. Prod. Commun. 2022, 17, doi:10.1177/1934578X211069721.
Bento, C.; Gonçalves, A.C.; Silva, B.; Silva, L.R. Peach (Prunus Persica): Phytochemicals and health benefits. Food Rev. Int. 2020, 1–32, doi:10.1080/87559129.2020.1837861.
Jesus, F.; Gonçalves, A.C.; Alves, G.; Silva, L.R. Health Benefits of Prunus avium plant parts: An unexplored source rich in phenolic compounds. Food Rev. Int. 2020, doi:10.1080/87559129.2020.1854781.
Ceniti, C.; Costanzo, N.; Morittu, V.M.; Tilocca, B.; Roncada, P.; Britti, D. Review: Colostrum as an emerging food: Nutraceutical properties and food supplement. Food Rev. Int. 2022, 00, 1–29, doi:10.1080/87559129.2022.2034165.
Erickson, M.D.; Yevtushenko, D.P.; Lu, Z.-X. Oxidation and thermal degradation of oil during frying: A review of natural antioxidant use. Food Rev. Int. 2022, 1–32, doi:10.1080/87559129.2022.2039689.
Fan, X.; Han, J.; Zhang, F.; Chen, W.; Fan, X. Red yeast rice: a functional food used to reduce hyperlipidemia Red yeast rice: a functional food used to reduce hyperlipidemia. Food Rev. Int. 2022, 1–27, doi:10.1080/87559129.2022.2043894.
Alam, A.; Al Arif Jahan, A.; Bari, M.S.; Khandokar, L.; Mahmud, M.H.; Junaid, M.; Chowdhury, M.S.; Khan, M.F.; Seidel, V.; Haque, M.A. Allium vegetables: Traditional uses, phytoconstituents, and beneficial effects in inflammation and cancer. Crit. Rev. Food Sci. Nutr. 2022, 1–35, doi:10.1080/10408398.2022.2036094.
Xu, S.; Liao, Y.; Wang, Q.; Liu, L.; Yang, W. Current studies and potential future research directions on biological effects and related mechanisms of allicin. Crit. Rev. Food Sci. Nutr. 2022, 1–27, doi:10.1080/10408398.2022.2049691.
Ohishi, T.; Hayakawa, S.; Miyoshi, N. Involvement of microRNA modifications in anticancer effects of major polyphenols from green tea, coffee, wine, and curry. Crit. Rev. Food Sci. Nutr. 2022, 1–32, doi:10.1080/10408398.2022.2038540.
Munekata, P.E.S.; Yilmaz, B.; Pateiro, M.; Kumar, M.; Domínguez, R.; Shariati, M.A.; Hano, C.; Lorenzo, J.M. Valorization of by-products from Prunus genus fruit processing: Opportunities and applications. Crit. Rev. Food Sci. Nutr. 2022, 0, 1–16, doi:10.1080/10408398.2022.2050350.
Taghian Dinani, S.; van der Goot, A.J. Challenges and solutions of extracting value-added ingredients from fruit and vegetable by-products: a review. Crit. Rev. Food Sci. Nutr. 2022, 0, 1–23, doi:10.1080/10408398.2022.2049692.
Fernández-Sobrino, R.L.; Torres-Fuentes, C.; Isabel Bravo, F.; Muguerza, B. Winery by-products as a valuable source for natural antihypertensive agents. Crit. Rev. Food Sci. Nutr. 2022, 1–14, doi:10.1080/10408398.2022.2049202.
Alrosan, M.; Tan, T.C.; Koh, W.Y.; Easa, A.M.; Gammoh, S.; Alu’datt, M.H. Overview of fermentation process: structure-function relationship on protein quality and non-nutritive compounds of plant-based proteins and carbohydrates. Crit. Rev. Food Sci. Nutr. 2022, 0, 1–15, doi:10.1080/10408398.2022.2049200.
Wei, Y.; Xu, J.; Miao, S.; Wei, K.; Peng, L.; Wang, Y.; Wei, X. Recent advances in the utilization of tea active ingredients to regulate sleep through neuroendocrine pathway, immune system and intestinal microbiota. Crit. Rev. Food Sci. Nutr. 2022, 0, 1–29, doi:10.1080/10408398.2022.2048291.
Pourjafar, H.; Ansari, F.; Sadeghi, A.; Samakkhah, S.A.; Jafari, S.M. Functional and health-promoting properties of probiotics’ exopolysaccharides; isolation, characterization, and applications in the food industry. Crit. Rev. Food Sci. Nutr. 2022, 1–32, doi:10.1080/10408398.2022.2047883.
Zhao, Y.; Li, M.; Wang, Y.; Geng, R.; Fang, J.; Liu, Q.; Kang, S.G.; Zeng, W.C.; Huang, K.; Tong, T. Understanding the mechanism underlying the anti-diabetic effect of dietary component: a focus on gut microbiota. Crit. Rev. Food Sci. Nutr. 2022, 1–21, doi:10.1080/10408398.2022.2045895.
Zayed, A.; Sheashea, M.; Kassem, I.A.A.; Farag, M.A. Red and white cabbages: An updated comparative review of bioactives, extraction methods, processing practices, and health benefits. Crit. Rev. Food Sci. Nutr. 2022, 1–18, doi:10.1080/10408398.2022.2040416.
Zhao, L.; Zhang, M.; Mujumdar, A.S.; Wang, H. Application of carbon dots in food preservation: a critical review for packaging enhancers and food preservatives. Crit. Rev. Food Sci. Nutr. 2022, 1–19, doi:10.1080/10408398.2022.2039896.
Lin, H.; Jiang, H.; Adade, S.Y.-S.S.; Kang, W.; Xue, Z.; Zareef, M.; Chen, Q. Overview of advanced technologies for volatile organic compounds measurement in food quality and safety. Crit. Rev. Food Sci. Nutr. 2022, 1–23, doi:10.1080/10408398.2022.2056573.
Monsalve-Atencio, R.; Montaño, D.F.; Contreras-Calderón, J. Molecular imprinting technology and poly (ionic liquid)s: Promising tools with industrial application for the removal of acrylamide and furanic compounds from coffee and other foods. Crit. Rev. Food Sci. Nutr. 2022, 1–20, doi:10.1080/10408398.2022.2038078.
Hanley-Cook, G.T.; Daly, A.J.; Remans, R.; Jones, A.D.; Murray, K.A.; Huybrechts, I.; De Baets, B.; Lachat, C. Food biodiversity: Quantifying the unquantifiable in human diets. Crit. Rev. Food Sci. Nutr. 2022, 1–15, doi:10.1080/10408398.2022.2051163.
Sun, H.; Yang, S.; Zhao, W.; Kong, Q.; Zhu, C.; Fu, X.; Zhang, F.; Liu, Z.; Zhan, Y.; Mou, H.; et al. Fucoxanthin from marine microalgae: A promising bioactive compound for industrial production and food application. Crit. Rev. Food Sci. Nutr. 2022, 1–17, doi:10.1080/10408398.2022.2054932.
Li, Z.; Tian, J.; Cheng, Z.; Teng, W.; Zhang, W.; Bao, Y.; Wang, Y.; Song, B.; Chen, Y.; Li, B. Hypoglycemic bioactivity of anthocyanins: A review on proposed targets and potential signaling pathways. Crit. Rev. Food Sci. Nutr. 2022, 1–18, doi:10.1080/10408398.2022.2055526.
Meruvu, H.; Harsa, S.T. Lactic acid bacteria: isolation–characterization approaches and industrial applications. Crit. Rev. Food Sci. Nutr. 2022, 1–20, doi:10.1080/10408398.2022.2054936.
Wang, M.; Zhou, J.; Tavares, J.; Pinto, C.A.; Saraiva, J.A.; Prieto, M.A.; Cao, H.; Xiao, J.; Simal-Gandara, J.; Barba, F.J. Applications of algae to obtain healthier meat products: A critical review on nutrients, acceptability and quality. Crit. Rev. Food Sci. Nutr. 2022, 1–18, doi:10.1080/10408398.2022.2054939.
Fan, X. Chemical inhibition of polyphenol oxidase and cut surface browning of fresh-cut apples fresh-cut apples. Crit. Rev. Food Sci. Nutr. 2022, 1–15, doi:10.1080/10408398.2022.2061413.
Sultana, S.; Bouyahya, A.; Rebezov, M.; Shariati, M.A.; Balahbib, A.; Khouchlaa, A.; Yaagoubi, O.M. El; Khaliq, A.; Omari, N. El; Bakrim, S.; et al. Impacts of nutritive and bioactive compounds on cancer development and therapy. Crit. Rev. Food Sci. Nutr. 2022, 1–30, doi:10.1080/10408398.2022.2062699.
Nathália, A.; Corrêa, R.; Ferreira, C.D. Essential oil for the control of fungi, bacteria, yeasts and viruses in food: an overview. Crit. Rev. Food Sci. Nutr. 2022, 1–15, doi:10.1080/10408398.2022.2062588.
Zhu, S.; Xu, J.; Adhikari, B.; Lv, W.; Chen, H. Nostoc sphaeroides Cyanobacteria: a review of its nutritional characteristics and processing technologies. Crit. Rev. Food Sci. Nutr. 2022, 1–17, doi:10.1080/10408398.2022.2063251.
Wu, J.; Cui, S.; Liu, J.; Tang, X.; Zhao, J.; Zhang, H.; Mao, B.; Chen, W. The recent advances of glucosinolates and their metabolites: Metabolism, physiological functions and potential application strategies. Crit. Rev. Food Sci. Nutr. 2022, 1–18, doi:10.1080/10408398.2022.2059441.
Yang, J.; Kuang, H.; Li, N.; Hamdy, A.M.; Song, J. The modulation and mechanism of probiotic-derived polysaccharide capsules on the immune response in allergic diseases. Crit. Rev. Food Sci. Nutr. 2022, 1–13, doi:10.1080/10408398.2022.2062294.
Singh, S.; Sharma, A.; Monga, V.; Bhatia, R. Compendium of naringenin: Potential sources, analytical aspects, chemistry, nutraceutical potentials and pharmacological profile. Crit. Rev. Food Sci. Nutr. 2022, 1–32, doi:10.1080/10408398.2022.2056726.
Liu, D.; Guo, Y.; Ma, H. Production, bioactivities and bioavailability of bioactive peptides derived from walnut origin by-products: a review. Crit. Rev. Food Sci. Nutr. 2022, 1–16, doi:10.1080/10408398.2022.2054933.
(and so on…)

Round 3
Reviewer 1 Report
The work improved consistently from the beginning today the review process. I thank the authors to have accepted my suggestion throughout the review process and I wish the best of luck for the best steps.
Reviewer 2 Report
The authors responded to all my comments.